# Taking a shortcut: what mechanisms do fish use?
Adelaide Sibeaux [1] ✉, Cait Newport[1], Jonathan P. Green[1], Cecilia Karlsson [2], Jacob Engelmann [3] & Theresa Burt de Perera[1]

Path integration is a powerful navigational mechanism whereby individuals continuously update their distance and angular vector of movement to calculate their position in relation to their departure location, allowing them to return along the most direct route even across unfamiliar terrain. While path integration has been investigated in several terrestrial animals, it has never been demonstrated in aquatic vertebrates, where movement occurs through volumetric space and sensory cues available for navigation are likely to differ substantially from those in terrestrial environments. By performing displacement experiments with *Lamprologus ocellatus*, we show evidence consistent with fish using path integration to navigate alongside other mechanisms (allothetic place cues and route recapitulation). These results indicate that the use of path integration is likely to be deeply rooted within the vertebrate phylogeny irrespective of the environment, and suggests that fish may possess a spatial encoding system that parallels that of mammals.

Path integration, sometimes called vector-based navigation, is a computational strategy whereby individuals continuously monitor their outbound distance and direction travel vectors, and integrate this information to produce a single "home" vector that takes them directly back to their point of origin[1]. Path integration allows animals to take the most direct route when homing, making it an energy-efficient method of navigating through any environment, whether novel or familiar[2,3]. This strategy may be of particular importance in spatially complex and dynamic habitats where landmark use can be challenging[4]. Path integration is underpinned by neural mechanisms that extract and integrate distance and direction information[5], and has been demonstrated in a number of terrestrial vertebrates (e.g., rodents, humans, geese) and insects[1,3,6–9]. In contrast, very little is known about the mechanisms underpinning navigation in animals that inhabit aquatic systems, such as fish. Aquatic environments are subject to very different constraints to those that are terrestrial, both in terms of the quantity of information available (aquatic environments are intrinsically volumetric) and quality of information (different sensory cues available). Despite the ubiquity of aquatic ecosystems, it is unknown whether path integration is universal in vertebrate clades, which would indicate that it is robust across very different environments and vertebrate groups.

Aquatic environments can be highly dynamic, and rapid fluctuations in conditions are likely to impact many sensory systems. For example, changing turbidity can introduce noise to olfactory and visual cues, and changes in water flow or tides can alter olfactory and hydrostatic cues. Path integration might therefore be a particularly important navigational strategy for aquatic species as the "home" vector can be updated without the use of external cues. Recent research has demonstrated that an aquatic invertebrate, the mantis shrimp, can navigate across a surface via path integration[10]. Crucially, however, we do not know whether path integration is also possible in aquatic vertebrates such as fish, which are nonsurface-bound animals and face a high complexity of navigation with six degrees of freedom of movement[11,12]. Additional problems for aquatic animals are that the flow of particles in water may impair the estimation of distance traveled based on visual cues such as optic flow[13,14], and that water current and tides can affect individual movements and swimming speed, thus compromising vector-based navigation. Testing whether teleost fish can path integrate will allow us to build a more cohesive picture of spatial navigation across different environments, and might also form a scaffold for research into the evolutionary origin of the neural mechanisms that underpin navigation.

To path integrate and to keep track of their translational and angular displacements (i.e., distance and direction), individuals need to extract sensory information that they can use as an odometer (i.e., idiothetic self-motion cues, e.g., vestibular, proprioceptive or optic flow cues) and compass (e.g., the sun, polarization pattern -allothetic compass- or angular self-motion -idiothetic compass-). Individuals must also possess the neural substrate to encode and integrate these sensory inputs. Crucially, they

[1]Department of Biology, University of Oxford, Zoology Research and Administration Building, 11a Mansfield Road, Oxford OX1 3SZ, UK. [2]Wolfson College, University of Cambridge, Cambridge CB3 9BB, UK. [3]Faculty of Biology, Bielefeld University, Universitätstrasse 25, Bielefeld 33615, Germany. ✉e-mail: adelaide.sibeaux@biology.ox.ac.uk

require an accumulator to encode the home-vector, which would be updated as the individual travels[15]. While a variety of cells that underpin spatial cognition have been found in the goldfish (e.g., speed cells, boundary vectors cells), the presence of a neural substrate for path integration in fish has not yet been identified[16–18]. Behavioral evidence of path integration will therefore not only highlight its existence as a navigational strategy in fish, but could also inform the exploration of the neural systems associated with this behavior.

Here, we present the first test of path integration in a teleost fish species, using the ostracophilic cichlid *Lamprologus ocellatus*. Males in this species defend a territory ($1–3$ m²) containing a variable number of snail (*Neothauma* sp.) shells; 1–2 shells are used by the male as a refuge from predators (hereafter, referred to as "home shells"), while any additional shells in the territory are used as spawning sites by females[19]. For males, the ability to path integrate and thereby keep track of their position relative to their home shell is expected to be advantageous, allowing them to return rapidly to the shell should they encounter a predator or competitor within the territory[20]. Their natural environment (Lake Tanganyika, Africa) is subject to variable turbidity levels[21], which might cause landmark-based navigation to be error-prone due to misidentification or increase of detectability distance. It can also lead to navigation at slower speed[22].

We developed a novel behavioral paradigm that takes advantage of the strong motivation of males to return as efficiently as possible to their home shells. A single individual was placed in a large experimental arena ($1.3$ m²) containing a shell the day before testing (Fig. 1). The arena was surrounded by a white, opaque sheet to prevent the fish from using visual cues outside the arena (including the presence of the experimenter). Above the arena, the sheet had a black and white checkerboard pattern which hid cameras placed in the black squares, and potentially provided optic flow cues. On the day of testing, an L-shape tunnel that terminated in a food reward chamber was

placed in the arena (Fig. 1). Once the fish reached the reward chamber, a guillotine door was closed, and the fish was contained. The tunnel and shell were removed from the arena and the chamber was displaced to a novel location (Fig. 1b, c). The chamber door was then opened and the swimming trajectory of the fish was recorded by an overhead camera. We compared observed trajectories with those predicted by the following three navigational strategies: (1) path integration (PI): the individual traveled in a straight-line in the direction where the home would have been (outward angle of 45°), ignoring the passive displacement[1]; (2) allothetic place cues (APC): the individual returned to the original position of the shell (outward angle of 135°), and (3) route recapitulation (RR): the individual retraced its previous trajectory along an L-shaped path that mirrored the shape of the tunnel.

Our results showed evidence consistent with cichlids using path integration to navigate, but also demonstrated that they use more than one navigational strategy, including the use of allothetic information and route recapitulation. Our data implies that the fish had integrated the trajectory from their home shell to the food reward based on the summation of the outward movement vector. This demonstrates that path integration is not limited to terrestrial animals, and that this ability is present in the early vertebrate lineage. Some fish returned to the original position of the shell, demonstrating the use of allothetic place cues (e.g., geometry of the square arena). Finally, evidence consistent with route recapitulation suggests that some fish also learned the outward journey as two consecutive vectors. A substantial proportion of individuals displayed random swimming trajectories, suggesting that they either did not learn any spatial information prior to displacement, or that they lacked motivation to return to their shell. The range of strategies observed is consistent with the hypothesis that navigational strategies are not mutually exclusive; it is likely that it is a combination of possible strategies that ensures navigation is accurate and robust to changing environmental conditions.

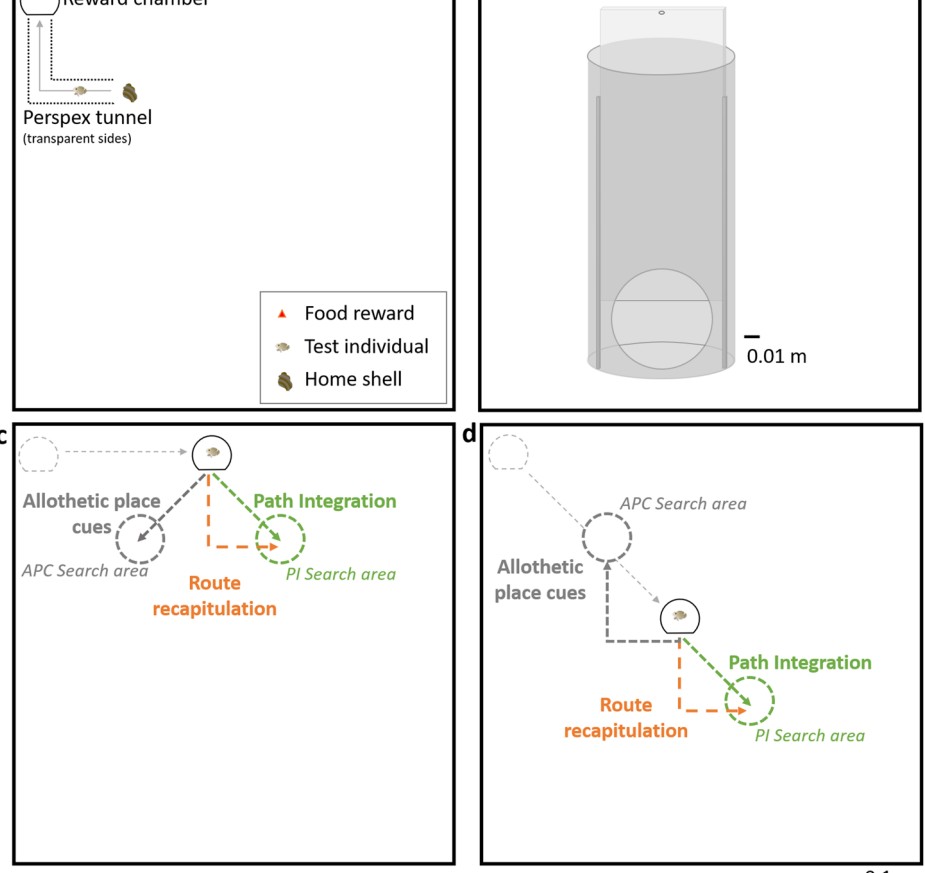

**Fig. 1 | Experimental overview for testing path integration in *Lamprologus ocellatus*. a** Following overnight acclimation in the experimental tank with a home shell, the fish was given the opportunity to reach a high-quality food reward (red triangle) by swimming through a Perspex L-shaped tunnel to a reward chamber. Once the fish reached the reward chamber the door was closed to contain the fish, and the chamber was displaced to a novel position. **b** Details of the reward chamber, 9.5 cm in diameter and 19.5 cm in height with a 6.5 cm diameter door. The door was attached to the inner walls of the chamber. **c** Lateral displacement (0.5 m to the right). **d** Diagonal displacement. (0.5 m to the right, 0.5 m down). After displacement, the L-shaped maze and shell were removed. The fish was then released from the reward chamber by opening the door and its movements were recorded with an overhead camera. Path integration strategy (dotted green arrow) and search area (dotted green circle) show the expected path and search areas if the fish uses path integration to locate its home shell. Allothetic place cues strategy (dotted gray arrow) and search area (dotted gray circle) show the expected path and search areas if the fish uses geometrical cues or global landmark cues (tank corners or distance to the wall) to locate its home shell. Route recapitulation strategy (dotted orange arrow) is expected if the fish reproduce outward path during its inward journey.

**Table 1 | Trajectories followed by the 24 cichlids released from the reward chamber after displacement**

| Fish ID | Model followed | | | | Number of times performing experiment |
|---|---|---|---|---|---|
| | PI | APC | RR | Random | |
| 1 | | | | 2 | 2 |
| 3 | | 1 | | 1 | 2 |
| 4 | | | | 2 | 2 |
| 5 | 1 | | | | 1 |
| 6 | | | 1 | 1 | 2 |
| 10 | | 1 | | | 1 |
| 12 | | | | 1 | 1 |
| 14 | | | 1 | | 1 |
| 15 | | | | 1 | 1 |
| 16 | 1 | | | | 1 |
| 18 | 1 | | | 1 | 2 |
| 19 | 1 | | | | 1 |
| 20 | | | 1 | | 1 |
| 21 | 1 | | | 1 | 2 |
| 22 | 1 | | | 1 | 2 |
| 25 | | | 1 | | 1 |
| 26 | | 1 | | 1 | 2 |
| 27 | | | | 1 | 1 |
| 29 | | 1 | | 1 | 2 |
| 31 | | 1 | | 1 | 2 |
| 33 | | 1 | | 1 | 2 |
| 34 | | | 1 | | 1 |
| 35 | 1 | | | 1 | 2 |
| 39 | 1 | | | 1 | 2 |
| Total | 8 | 6 | 5 | 18 | 37 |
| Subtotal Lateral | 6 | 4 | 2 | 9 | |
| Subtotal Diagonal | 2 | 2 | 3 | 11 | |

*PI* Path integration, *APC* Allothetic place cues, *RR* Route recapitulation. Light blue values indicated the trajectory was followed in the first trial. Dark blue values indicated that the trajectory was followed in the second trial (regardless of whether the first trial was successful). The subtotal values indicate the number of fish that were under lateral or diagonal displacement.

## Results

We tested 40 fish over 80 trials (once in each of the lateral or diagonal displacement configuration, Fig. 1b, c). Of those, 24 individuals entered and exited the reward chamber, resulting in a total of 37 trials which could be analysed (only 13 individuals exited the chamber in the two displacement configurations). In a quarter of the trials ($n = 19$) the fish stayed close to their shell and never entered the second branch of the L-shape tunnel, in two trials the fish never left their shells, the relatively low engagement rate highlights the attachment of male *L. ocellatus* to their shells, which was a prerequisite for testing path integration.

Individuals did not display any search behavior, such as loops of ever-increasing size around the expected location of the starting point[23]. Therefore, each trajectory, beginning at the opening of the reward chamber, was cropped once the fish oriented back to the chamber, following e.g., ref. 24 (see methods and Supplementary Fig. S1 for details). A circular statistical analysis ("CircMLE" model-based approach[25]) on the trajectories' orientation allowed us to demonstrate that our data fit multiple modal direction distribution significantly better than uniform or unimodal direction and therefore grouping could be applied (rt statistics = 0.56, $p < 0.001$, $\Delta$AIC = 0, see methods and Supplementary Table S2a and S2b for details). Route similarity, orientation angle and distance traveled of the observed trajectories were then compared to trajectories predicted for Path integration (PI), Allothetic place cues (APC) and Route Recapitulation (RR) strategies.

### Route similarity

First, 1000 evenly spaced points were fitted to the fish's observed trajectory and each of the model trajectories (PI, APC, RR) both cropped to the same size. Then, the Euclidian distance between each pair of interpolated points

was extracted (i.e., distance between the tenth interpolated point in the trajectory from the tenth interpolated point in the model trajectory) and used to test if the fish's trajectory was significantly closer to one of the three model trajectories: PI, APC or RR (Linear mixed model, Supplementary Table S3, see methods for details). The average distance between the fish trajectory and its closest model trajectory was then compared to 10,000 randomly generated trajectories to determine whether the fish's trajectory was closer to the model trajectory or non-significantly different from random.

Out of 37 trajectories, 8 followed a path integration trajectory (PI, Table 1, Figs. 2, 3), 6 returned to the original position of the shell, following an allothetic place cue trajectory (APC, Table 1, Figs. 2, 3), 5 retraced the outbound path following a route recapitulation trajectory (RR, Table 1, Figs. 2, 3), and 16 were not significantly different from random (Table 1).

### Orientation angle

Splitting the trajectories into the following groupings (PI: 8 trajectories; APC: 6 trajectories; RR: 5 trajectories), we tested if the orientation angle (first angle and angle start-end) matched the angle predicted by each strategy. The first angle refers to the fish's first orientation when exiting the reward chamber. The angle start-end refers to the angle between the first point out of the chamber and the last point of the fish's trajectory. We did not find any significant differences between the model expectation (45° angle) and the fish trajectories following the PI strategy (Table 2, Fig. 2) and for the fish following APC strategy after lateral displacement (expectation: 135° angle, Table 2, Fig. 2). Only two fish followed the APC strategy after diagonal displacement, precluding statistical analysis (Table 2, Fig. 2). For the fish grouped under the RR strategy, we did not find any significant

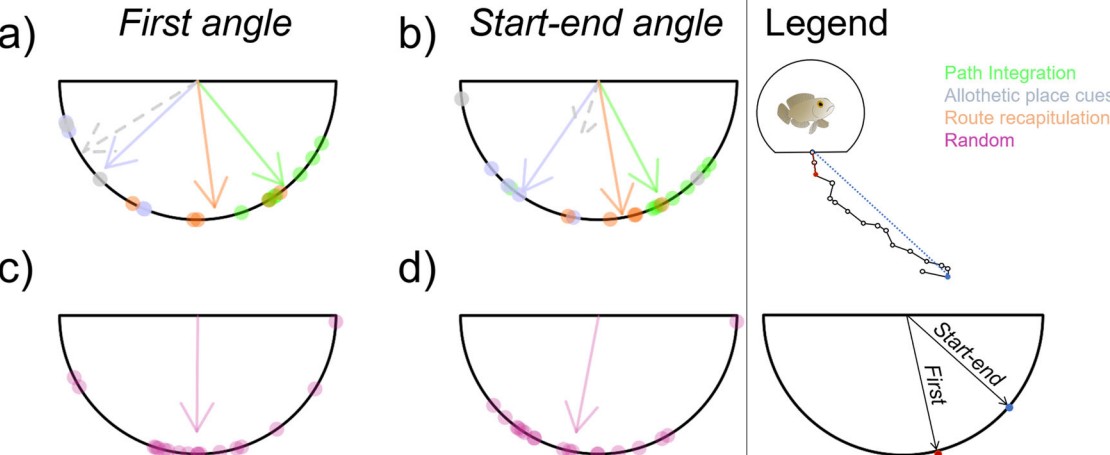

**Fig. 2 | Fish navigation angles.** First angle (**a**, **c**) and angle start-end (**b,d**) taken by the fish following Path Integration (green), Allothetic place cues (gray), Route recapitulation (orange) strategies and for the fish showing a random trajectory (purple, **c** and **d**). The fish following APC after lateral or diagonal displacement are represented by the plain and dashed gray line respectively. The direction of the arrow represents the average orientation of each fish group and its length represents the strength of the uniformity of the direction taken by all fish in the group (Rayleigh test, circular statistics, see methods).

differences between the model expectation and the fish's first angle; however, the angle start-end was significantly different from the model expectation (Table 2, Fig. 2).

### Distance traveled

We tested whether observed distance traveled by the fish grouped under the three model trajectories (PI, APC, RR) matched distance expected by the model (see methods for details).

We did not find any significant differences in distance traveled between the model expectation (38.6 cm) and the fish trajectories following the PI strategy (Table 2, Fig. 3) and for the fish following APC strategy after lateral displacement (expectation: 38.6 cm, Table 2, Fig. 3). We found a significant difference between the model expected distance traveled (54.5 cm) and the fish following the RR strategy (Table 2, Fig. 3). For RR, the difference in the angle start-end and distance traveled from the model expectation (Table 2, Fig. 2, Fig. 3) can be explained by the fact that the fish only retraced the first part of the L-shaped tunnel before orienting back toward the reward chamber. Moreover, the average distance traveled was not significantly different amongst the PI, RR, ACP, and Random orientation groups except for the RR and random orientation groups ($\chi^2 = 10.05$, df = 4, $p = 0.040$, and a posthoc pairwise Wilcoxon test indicates $p = 0.049$ with Holm–Bonferroni correction, Fig. 4, see Supplementary Table S5 for average path length and straightness for each group).

### Effect of age, swimming speed, test trial number, and displacement on chosen strategy

Using a multinomial mixed model, we tested whether extrinsic factors (displacement: Lateral versus Diagonal, trial number: first or second test) and intrinsic (fish age, swimming speed) affected an individual's propensity to follow a PI, APC or RR strategy or to show random swimming behavior. Strategy (PI, APC, RR or random) was added as the multinomial response variable. Fish age, swimming speed, displacement, and trial number were added as fixed effects, and fish identity as a random intercept. Sample sizes for this analysis were small and caution is required when interpreting the results. Diagonal displacement significantly affected the probability of using one of the strategies: fish displaced diagonally were more likely to follow a random trajectory over PI, APC or RR (Supplementary Table S6). Then, where an individual completed two trials, the first trajectory was more likely to be a random trajectory than PI (seven out of eight PI trajectories were performed the second time the fish was in the experimental tank, Table 1, Supplementary Table S6). Finally, individuals' age and swimming speed significantly predicted their navigation strategy: younger fish and slower

swimmers were more likely to use PI and RR trajectories over random movement (Supplementary Table S6).

### Discussion

Our results demonstrate, that a teleost species can follow a homing trajectory consistent with path integration, and by extension are likely to be able to encode and integrate distance and direction vectors. This is the second time vector-based navigation has been observed in non-terrestrial species[26] and the first time in an aquatic vertebrate. In fact, there are clues that suggest that path integration may be widespread among teleost fish. One requirement of path integration is that the navigator can estimate distance traveled. This ability was recently demonstrated in the marine coral reef Picasso triggerfish[13], and freshwater goldfish[14]. These two species, phylogenetically distant, inhabit different environments in terms of the types of allothetic place cues available, water movement, and propagation of sensory signals. In our study, local landmark cues were removed or concealed, ensuring that distance traveled estimates were generated by the cichlids own self-motion. For example, the cichlid's odometer could have been based on the optic flow cues generated as they moved over the sandy bottom, or by proprioceptive cues such as their number of fin beats. These combined results suggest that distance estimation is highly conserved in fish, and not dependent on environmental characteristics.

Our results indicate that path integration is not the only navigation strategy that cichlids use, even under identical experimental conditions. In fact, we found evidence of all three types of strategies we tested (i.e., path integration, use of allothetic place cues and route recapitulation). The use of these different strategies indicates that *L. ocellatus* is able to encode both allothetic (e.g., boundaries, environmental geometry) and idiothetic cues (e.g., self-generated distance and direction vectors). Exploring the interaction between these different navigational strategies, and understanding whether they are complementary, interchangeable, or hierarchal in nature, represents important directions for future research.

Inter- and intra-individual variation in navigational approaches is commonly observed across taxa (e.g., fish[27-30], mantis shrimps[26], ants[31,32], or humans[33]). In a spatial orientation experiment, goldfish used both visual features (visual pattern displayed on a wall) and geometric cues (corner of the rectangular testing apparatus) to orient, with equal preferences for each when cues were in conflict[34]. Similarly, mantis shrimps showed individual variation in their homeward navigation when landmark cues and path integration were placed in conflict[26]. Rather than being a rigid behavior, an individual's navigational strategy appears to be context-dependent, with the choice of strategy dependent in part on the reliability of cues present in the environment[29,32]. For

**Fig. 3 | Individual swimming trajectories.** The individual swimming trajectories are displayed for the fish following PI (green), APC (gray) or RR (orange) strategies and for the fish showing a random trajectory (purple). Trajectories that followed a lateral displacement are presented as plain line, trajectories that followed a diagonal displacement are presented as dashed lines. The black lines represent the model trajectories. **a** Individual trajectories cropped to the size of the model trajectories, (**b**) full individual trajectories. Full trajectories were used to test if the observed distanced matched distance expected by the model (Table 2).

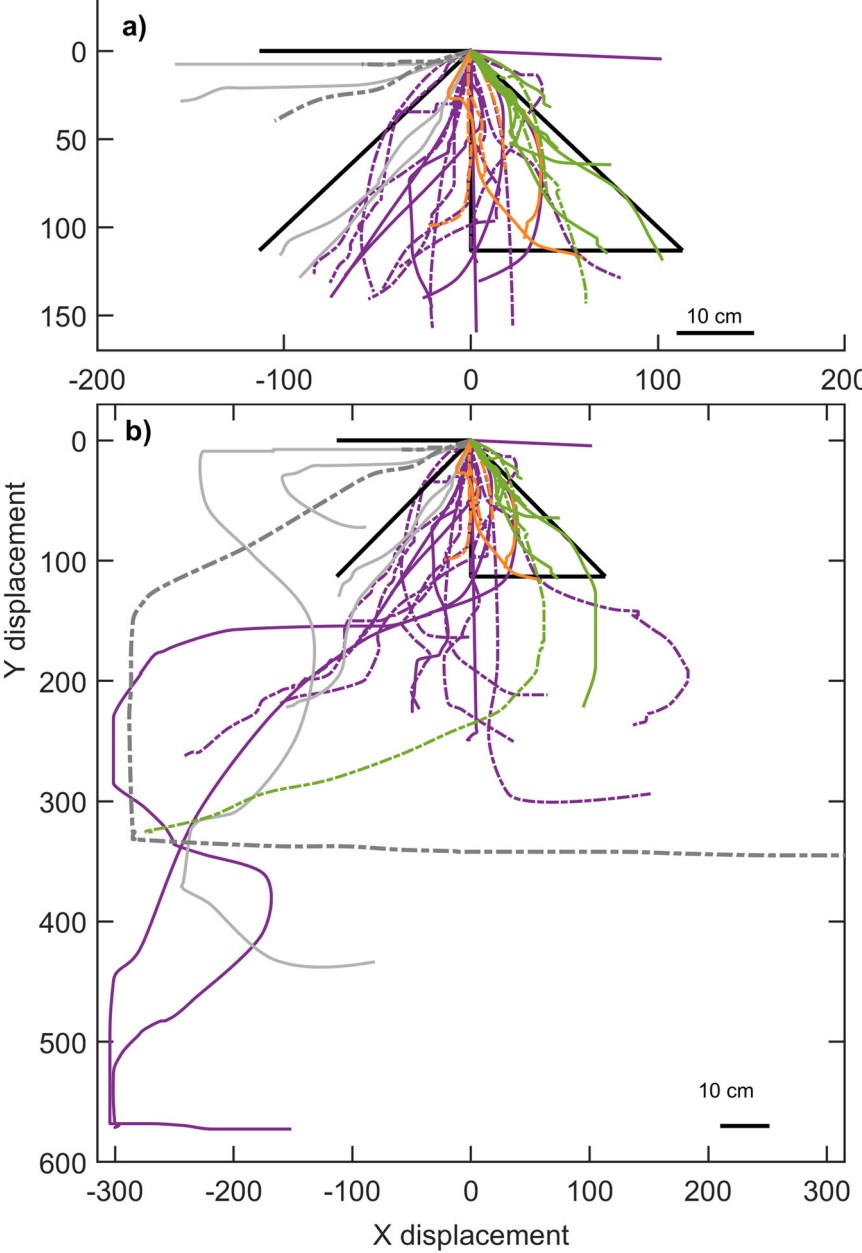

example, ants found in featureless habitats are more likely to use path integration than those inhabiting structurally complex environments where visual landmarks are available[31]. Similarly, while humans can path integrate, they are more likely to use landmarks to navigate in stable environments[33].

In our experiment, some of the behavioral variability could be explained by intrinsic factors (individual age and swimming speed) and extrinsic design factors (e.g., trial number or displacement in the tank). It is also possible that individuals have preferences for different types of navigational strategies. Of the 13 fish tested twice, we did not observe any individuals changing strategies (i.e., PI, AC or RR): where one trial differed from another, one trajectory was always categorized as random. Thus, it is possible that individual navigational strategies are fixed within individuals. Perhaps more likely, the presence of landscape cues may have led to difference in navigational approaches. Individuals following an allothetic place cue strategy rely on their ability to detect, encode, and use information from external sensory cues. Our experimental design controlled for the use of most allothetic place cues. For example, the shell or tunnel were removed, prohibiting their use as local landmarks. Global landmarks from the laboratory were concealed by a curtain, and odour cues were scrambled by

stirring the sand and water after each displacement. However, some allothetic place cues remained, which may have influenced the navigational strategy the fish used. For example, the corner of the square tank could provide geometric cues. In a spontaneous reorientation task (i.e., where no reinforced training was used), ref. 35. showed that geometry cues of a rectangle arena were used by two different fish species to orient towards the previous position of a conspecific. A neurophysiological study by ref. 16. revealed the presence of edge detection neurones in the lateral pallium of teleost fish, which is a first indicator that fish are able to encode geometric information. We therefore recommend the use of a circular arena in further experiments. In our experiment, cues may have also been provided by the checkerboard pattern suspended above the water surface. Our estimate of the visual acuity of *L. ocellatus* (4.40 CPD, see methods) suggests that they may have been able to see details of the highly contrasted checkerboard pattern 1.5 m above the tank (see Fig. 1 in ref. 36. for visual prediction a low CPD). While *L. ocellatus* inhabits depths of up to 20 m in the wild[19], it is possible that in our experimental aquaria, with 0.18 m of water and a depth of 0.45 m (distance to the checkerboard pattern), individuals may have attended to visual information above the water surface. A recent study by

**Table 2 | Differences between travel metrics expected by the model trajectories and fish trajectories**

| Fish group | Travel metric | t | df | V | p | mean | 95% CI | Model expectation |
|---|---|---|---|---|---|---|---|---|
| PI (n = 8) | First straight angle | 1.03 | 7 | - | 0.356 | 50.44 | [40.51 ; 59.64] | 45° |
| | Angle start-end | - | - | 32 | 0.055 | 63.9 | [43.92 ; 83.87] | 45° |
| | Distance traveled | - | - | 14 | 0.641 | 41.81 | [13.14 ; 70.49] | 38.6 cm |
| APC Lateral (n = 4) | First straight angle | 0.14 | 3 | - | 0.622 | 136.94 | [112.91 ; 160.88] | 135° |
| | Angle start-end | −1.23 | 3 | - | 0.240 | 124.4 | [108.21 ; 136.93] | 135° |
| | Distance traveled | - | - | 10 | 0.125 | 94.24 | [20.69 ; 167.78] | 38.6 cm |
| APC Diagonal (n = 2) | First straight angle | two observations only: 135.00 and 161.56 | | | | | | 180° |
| | Angle start-end | two observations only: 44.38 and 172.69 | | | | | | 180° |
| | Distance traveled | two observations only: 14.65 and 277.16 | | | | | | 27.3 cm |
| RR (n = 5) | First straight angle | −0.64 | 4 | - | 0.498 | 82.35 | [62.21 ; 101.92] | 90° |
| | Angle start-end | 5.23 | 4 | - | **0.005** | 80.05 | [69.76 ; 92.33] | 45° |
| | Distance traveled | −7.48 | 4 | - | **0.023** | 25.62 | [19.17 ; 32.02] | 54.5 cm |

Results from bootstrap t test (t, df, bootstrapped p values, bootstrapped mean and 95% CI are provided) or Wilcoxon test (V, p value, mean and 95% CI are provided) depending on the data normality (see methods). Each line presents the result from a single test. The mean and 95% CI angles values are in degrees and the distance in cm. Significant p values are in bold.
PI Path integration, APC Allothetic place cues, RR Route recapitulation.

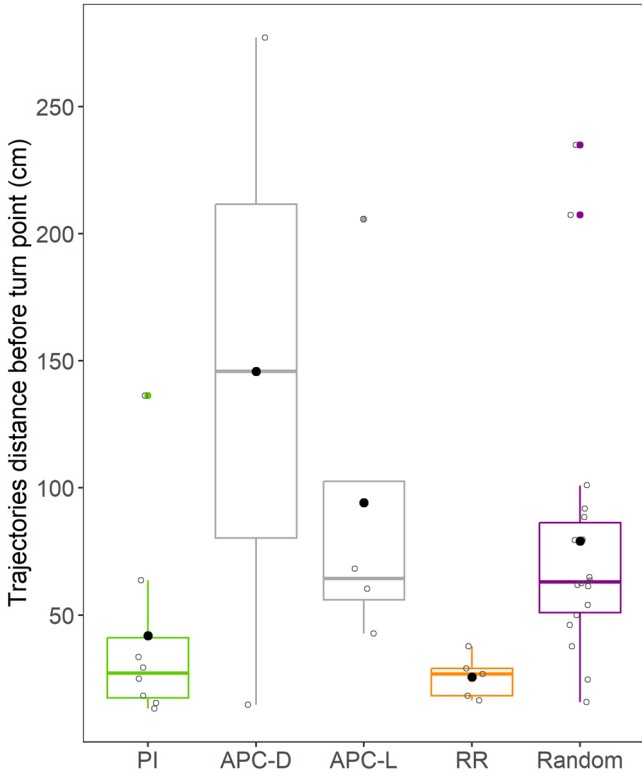

**Fig. 4 | Swimming distance for the fish following PI, APC, RR or random orientation.** Box plots indicate the median and interquartile range. Black dots represent the average distance for each group. Colored-filled circles indicate outliers. Empty circles represent individual distances of the fish full trajectory (before the fish orientates back to the reward chamber).

ref. 37. showed that redtail splitfins were able to integrate geometry and landmark cues under both spontaneous choice and training condition. Because no landmarks were present, our results suggest that *L. ocellatus* likely used the geometry of the experimental tank as a navigational cue that allowed them to return to the initial position of their shells. Given that other potential navigational cues were present, the variety of navigational behaviors seen among our experimental subjects is not surprising. Individual variability likely contributes to survival by allowing individuals and populations to adapt to ever-changing environmental conditions[28].

A few individuals followed a route recapitulation strategy, retracing their previous route. This suggests that these fish had memorized the path taken from their home shell to the food reward, and encoded it as two consecutive vectors. There are some potential benefits to this strategy in the wild. While a shortcut home may be faster and more energy efficient, any risks associated with the new path, such as the presence of a predator or an unpassable barrier, are unknown. However, comparison of the observed and predicted route recapitulation trajectories indicated that individuals only retraced the first vector before turning back toward the reward chamber. Therefore, at least in our experiments, this strategy may require additional cues (e.g., a beacon or landmark), or learning opportunities, to be effective.

Of the 37 trajectories we analysed, almost half were best described by random orientation. Our analysis was highly conservative as we wanted to be certain that the tested individuals were following one of the model trajectories without doubts. Therefore, it is conceivable that some trajectories were grouped as random movements when they should belong to one of the model trajectories. Examples of data from the path integrating animals[1] show that error in path integration movement can lead to variance in datasets often spanning over 45°. Data from these animals using path integration in isolation are rarely as acute as path integrating fish reported in our experiment. However, even though we could have missed a few individuals performing one of the model trajectories, this study aims to highlight the possibility of path integration strategy in fish and the diversity of navigational strategies employed. Following our analysis results, there are several possible explanations for the important number of individuals following random orientation. Some individuals, for example, may have decided to explore their environment and were not motivated to return directly to their shell on a particular trial. Alternatively, some individuals might priorities beacon or local landmark to navigate and could be lost and show random trajectories in their absence. Finally, individuals may have been stressed from the displacement procedure or isolation in the reward chamber. Stress has been previously shown to impact the spatio-cognitive abilities of fish (see Sandi[38] for a review on the effect of stress on cognitive performance). Fish may have also been disoriented by finding themselves in a different position than they had been before the door was shut. If they perceived the new scene as an entirely new location, we would not necessarily expect individuals to search for a shell associated with a different visual scene. Incidentally, fish displaced diagonally experienced a bigger change in scenery, and were more likely to display random movement. Although it is not possible to determine the cause of this behavior with certainty, seemingly randomly oriented trajectories are not surprising in animal experiments where there is no conditioning, and therefore the specifics of the task cannot be explained to the experimental subject.

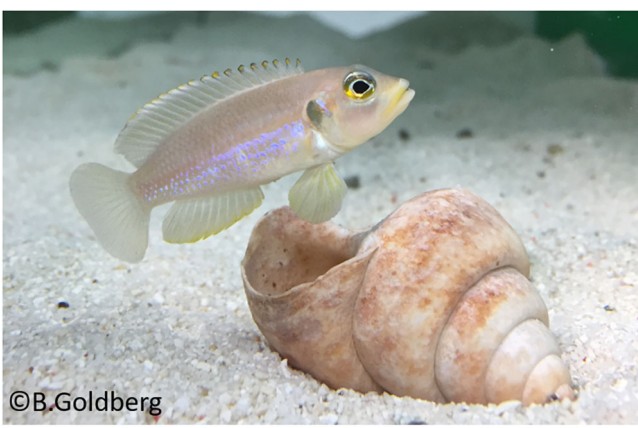

**Fig. 5 | *Lamprologus ocellatus*.** Male *Lamprologus ocellatus*, housed in the fish lab, Wytham, The University of Oxford. Credit: Dr Becca Goldberg.

The neural mechanisms and circuits associated with path integration in fish have yet to be identified, but our results that are consistent with path integration in fish indicate that *L. occelatus* must have the neural capacity to encode distance and direction vectors, and combine these two pieces of information to continuously calculate an efficient return route ("home" vector). Whether the mechanisms in fish are similar to that of mammals is an open question. In mammals, place cells and grid cells found in the hippocampal formation appear to be the neural substrate of path integration[5,39,40]. In teleost fish, a variety of cells used for spatial cognition have been recently recorded in the goldfish telencephalon (e.g., neural cells encoding head direction, speed, velocity, boundary vector cells[16,18]). Cerebellar circuit that integrates self-motion and stores self-location information have also been identified recently in zebrafish larvae. However, these cells allowed the fish to maintain a fixed spatial position in flowing water and it seems unlikely that maintaining positional equilibrium and path integration are controlled by the same neural circuits or that one mechanism underpins both tasks[17]. The neural substrate for path integration in teleost fish is still to be determined[16].

To conclude, while this study is consistent with the ability to path integrate in individual teleost fish, the evidence is not definitive and further experimental validation is needed. Nevertheless, the data suggests that vector-based navigation might be a common strategy used across animal taxa and within a wide range of habitats, including those that are aquatic. These results can form a scaffold to further experiments, and to neurophysiological studies that aim to unravel the mechanisms behind this ability, seeking to fully understand the representation of space across animal groups that inhabit very different environments.

## Methods
### Ethics and approval
The study was approved by the University of Oxford's Animal Welfare and Ethical Review Body (AWERB) (project code: APA/1/5/ZOO/NASPA/Burt/DistanceEstimation). We have complied with all relevant ethical regulations for animal use.

### Animal husbandry
We used 40 naive adult male cichlids (*Lamprologus ocellatus*, Fig. 5), sourced from a captive-bred laboratory population (standard body length = 3–5 cm). Individuals came originally from 6 subpopulations: B ($n = 4$), C ($n = 7$), D ($n = 1$), E ($n = 4$), F ($n = 16$), G ($n = 8$, Supplementary Table S4). Subjects were housed individually in tanks measuring 35 cm × 32 cm × 60 cm (width × height × length) containing a sand substrate to a depth of 4 cm and a single home shell. Illumination via fluorescent light followed a 12 h light/dark cycle. Individuals were fed twice a day, in the morning and the afternoon, with commercial flake food, supplemented with *Mysis* shrimps once a week. The tanks were cleaned once a week and the

water quality checked to maintain constant pH, gH and KH of 8.4, 23 and 12 ppm respectively. Ammonia and nitrites were kept at 0 ppm, while nitrates were maintained below 10 ppm. The water temperature was maintained at 26 ± 1 °C.

One individual (C39d), from subpopulation G, died between its two test sessions from an unknown cause and was replaced by individual C39 (Supplementary Table S1).

### Experimental apparatus
We used a 1.3 m² experimental tank ($1.3 \times 1.3 \times 0.45$ width, length and height respectively), with 0.06 m of sand substrate and 0.18 m of water. It was illuminated via fluorescent light following a 12 h light/dark cycle. Water parameters and water changes were managed as for the home tanks. On the top left corner of the experimental tank, a shell was placed in front of an L-shaped tunnel 0.6 m long (the two external walls were 0.3 m and 0.3 m, Fig. 1) and 0.07 m diameter. The 0.6 m length of the tunnel was chosen because it produced an average beeline distance in the center of the tunnel of 0.53 m, which is within the average range of inter-shell distance for this species[19]. The tunnel led to a transparent holding chamber of diameter 0.1 m where a food reward was placed. Later, in tests, before displacement, the transparent holding chamber will be covered by an opaque sheet to prevent the fish from observing its translocation in the tank or the experimenter. The holding chamber had a sliding door connected to the L-shaped tunnel. The sliding door was operated by the experimenter, situated outside the experimental apparatus. The sides of the experimental tank were covered by white opaque Perspex (31 cm in height). A plain white curtain (length: 6 m; height: 1.5 m) covered the entire tank to eliminate external visual cues. An achromatic checkerboard pattern printed on fabric was suspended above the experimental tank 1.5 m². Two holes (1 cm diameter) were cut in two different black squares of the checker pattern and cameras was mounted in these positions. The first camera was positioned above the reward chamber (top left corner of the experimental tank) and was used to visually ensure that the fish entered the chamber before the door was closed. The second camera was positioned in the center of the experimental tank and used to record the trajectories of the fish.

### Testing procedure
Individual fish were placed in the experimental tank for acclimation the day before the test (between 15:00 and 17:00). They were transferred from their home tank to the experimental apparatus within their home shell to limit their stress (see Fig. 1a for position in the experimental tank). The following morning, the L-shaped tunnel and chamber were placed into the experimental tank. A high-quality food reward (bloodworm) was placed in the chamber and the sliding door was then opened. Fish eventually swam into the reward chamber in search of food. Only once the fish reached the reward chamber and began eating, was the door shut. A black plastic sheet was then clipped onto the chamber to prevent the fish from seeing the experimenter. Once the tunnel and shell were removed, the holding chamber was slowly displaced either laterally: 0.5 m to the right (Fig. 1c) or diagonally: 0.71 m right-down (Fig. 1d). Those two displacement positions were chosen to ensure that the predicted search areas were at least 0.5 m from any tank walls, and reducing the likelihood that fish used the walls as navigational cues. The two positions were marked by white Perspex squares fixed to the bottom of the experimental tank and hidden under the sand. To ensure the chamber was displaced to an exact position, the experimenter aligned the Perspex squares with the chamber. After displacement, the sand substrate was mixed thoroughly and relevelled to scramble any visual or odour cues. The plastic sheet covering the reward chamber was then removed and the door was opened (mean ± sd latency before opening the reward chamber after displacement = 210 ± 50 s). Following release, the fish was allowed to swim in the experimental tank for 5 min. If the fish did not exit the reward chamber after 15 min, the trial was terminated.

Subjects were divided into two groups of 20 individuals (Supplementary Table S1). Each individual was tested in a lateral and a diagonal displacement test (Fig. 1b, c), with an interval of two weeks between tests.

Group 1 was tested first for lateral displacement and then for diagonal displacement and group 2 was tested for displacement diagonal and then lateral (see Supplementary Table S1).

## Data collection

The water temperature and times were recorded on both the acclimation and experiment day. The number of times each fish swam backwards and forward into the first and second branch of the L-shape tunnel before entering the holding chamber was recorded. The time taken to (1) enter the tunnel, (2) enter the chamber, and (3) exit the chamber after displacement was also recorded.

All trials were recorded using an overhead web camera (ELP Webcam 10-megapixel, Model X000VD0KT5). Videos were manually processed using custom code written for MATLAB version R2022a, MathWorks Inc. Every second, the position of the fish was identified by clicking on the head of the fish. X and Y coordinates were then extracted and used to produce a continuous 2D trajectory. Fish movements were analysed for one minute after exiting the chamber. Individuals did not display any search behavior, such as loops of ever-increasing size around the expected location of the starting point[23]. Therefore, each trajectory, beginning at the opening of the reward chamber, was cropped once the fish oriented back to the chamber, following e.g., ref. 24 (Supplementary Fig. S1).

Of the 40 fish tested, 24 completed one or both trials successfully ($n = 37$ successful trials). In the remaining trials, fish failed to enter the reward chamber (21 individuals, 26 trials), exit the reward chamber after displacement (6 individuals, 7 trials) or did not move away from the reward chamber after exiting (10 individuals, 10 trials).

## Statistics and reproducibility

Trajectory analysis was performed using custom MATLAB code. All statistical analyses were performed using R studio (version 2022.07.1, R version 4.0.2). The significance threshold was set at .05. All data and codes are shared through Dryad open data repository[41].

**Multimodal trajectories.** We used a model-based approach with maximum likelihood to investigate circular orientation. The CircMLE package[25] was used to determine if our trajectories followed uniform, unimodal or multimodal model orientation (for further details see ref. 25). We first used the package circular[42] to obtain "circular" class data with angles in radians and modulo = π, as our data could spread only from 0 to 180°. We used the "check_data" function from the CircMLE package to ensure our data were in the correct format and ran the 'circ_mle' function to test our orientation data including (a) first orientation data and (b) start to the end of the trajectory orientation data. The first orientation data followed model M5A (Homogenous bimodal) and the start-end orientation data followed M5B (Bimodal) models (see Supplementary Tables S2a and S2b for full details). This analysis indicates that multimodal orientation defines our data significantly better than uniform or unimodal. Grouping our data following navigational model trajectories can then be applied. We computed the circular mean direction[43] as a descriptive statistic and performed the following analyses on angles using ordinary statistics that are robust and powerful as our angles are restricted to 0–180° .

**Deviation from the model trajectories.** To measure the extent to which observed trajectories deviated from the three model trajectories (PI, APC and RR) prediction, we measured the average distance between both trajectories: We first cropped the smallest of either the model trajectory or the observed trajectory so that they were both of equal length. We then fitted the observed data using 1000 evenly spaced points using a modified akima interpolation (makima, MATLAB) and did the same with the model trajectories. Finally, we calculated the Euclidian distance between each pair of interpolated points (i.e., distance between the tenth interpolated point in the trajectory from the tenth interpolated point in the model trajectory). Average distances between the observed and the model

trajectories are presented in the Supplementary Table S3. All interpolated trajectories are presented in the Supplementary Fig. S2.

**Determination of the closest model trajectory.** A generalized mixed linear model (GLMM) was used to test the similarity between observed trajectories and the three model trajectories. We used glmmTBM (glmmTBM package[44]), which fits models using maximum likelihood estimation via "TMB" (Template Model Builder). For each fish, the distances between the 1000 points interpolated along the observed trajectory and the 1000 points interpolated model trajectories were added as the response variable. Model trajectory (i.e., PI, APC, RR) was added as the explanatory variable and as a random intercept. Distribution and dispersion of residuals were assessed using the DHARMa package[42]. No overdispersion of residuals was detected for any of the fish tested ($p > 0.05$). A well-fitted statistical model predicts normality of the residuals and homoscedasticity, but this was rarely observed in our models. The lack of residual normality was due to both the large number of samples ($n = 1000$) and inclusion of "model" as a categorial fixed effect. A large number of samples often leads to statistical significance even if deviations are minor. The lack of homoscedasticity can be explained by the difference in shape of the three model trajectories: two trajectories (PI, APC) are straight, while the third (RR) includes a right angle. Therefore, if the fish trajectory closely follows PI or APC we can expect a higher variance to the right-angle trajectory. While these issues prevent obtaining a perfect fit between the models and the data, the models we selected provided a better fit than alternatives, determined by comparison of 10 model structures using different GLMM packages, distributions and data transformations. Average distance ±SD between the observed fish trajectory and the model trajectories and Average distance ± IC$_{95\%}$ are presented in the Supplementary Table S3 and Supplementary Fig. S3 respectively. For further details, see raw data and model output in R Code-Dryad depository.

**Deviation from random trajectories.** To determine whether an individual's trajectory was closer to a random trajectory than to one of the three model trajectories (PI, APC, RR), we measured the distance between the observed trajectory and 10,000 randomly generated trajectories. Each random trajectory was a straight line of a given length and angle and was generated using a custom MATLAB code. The angle, between 0 and 180°, was randomly-generated on the MATLAB code. The length and was the same as the smaller of either the individual's trajectory or the PI model trajectory. We then measured the distance between the 1000 interpolated points of the randomly generated trajectories and the 1000 interpolated to the observed trajectory. We did this for 10,000 randomly generated trajectories. We then calculated the average distance between the observed trajectory and the 10,000 random trajectories. The distribution of the average distance between the observed trajectory and the 10,000 randomly generated trajectories was assessed using a Cullen and Frey graphic analysis from the "descdist" function (fitdistrplus package[45], Rstudio version 2022.12.0). It followed a continuous uniform distribution (See Supplementary Fig. S4). The standard deviation around the mean for uniform distribution is calculated as follows:

$$SD_{uniform} = \frac{(b - a)}{\sqrt{12}}$$

where $a$ and $b$ is the interval over which the continuous uniform distribution is defined.

If the difference between the observed trajectory and one of the model trajectories was less than the average distance minus SD$_{uniform}$ of all random trajectories, then the fish was assigned to the model trajectory category (Fig. 6a). If the average distance minus SD$_{uniform}$ from all random trajectories was less than the distance between the fish trajectory and the model trajectories, then the fish was assigned to the random trajectory category (Fig. 6c). If the standard deviation was calculated assuming a normal

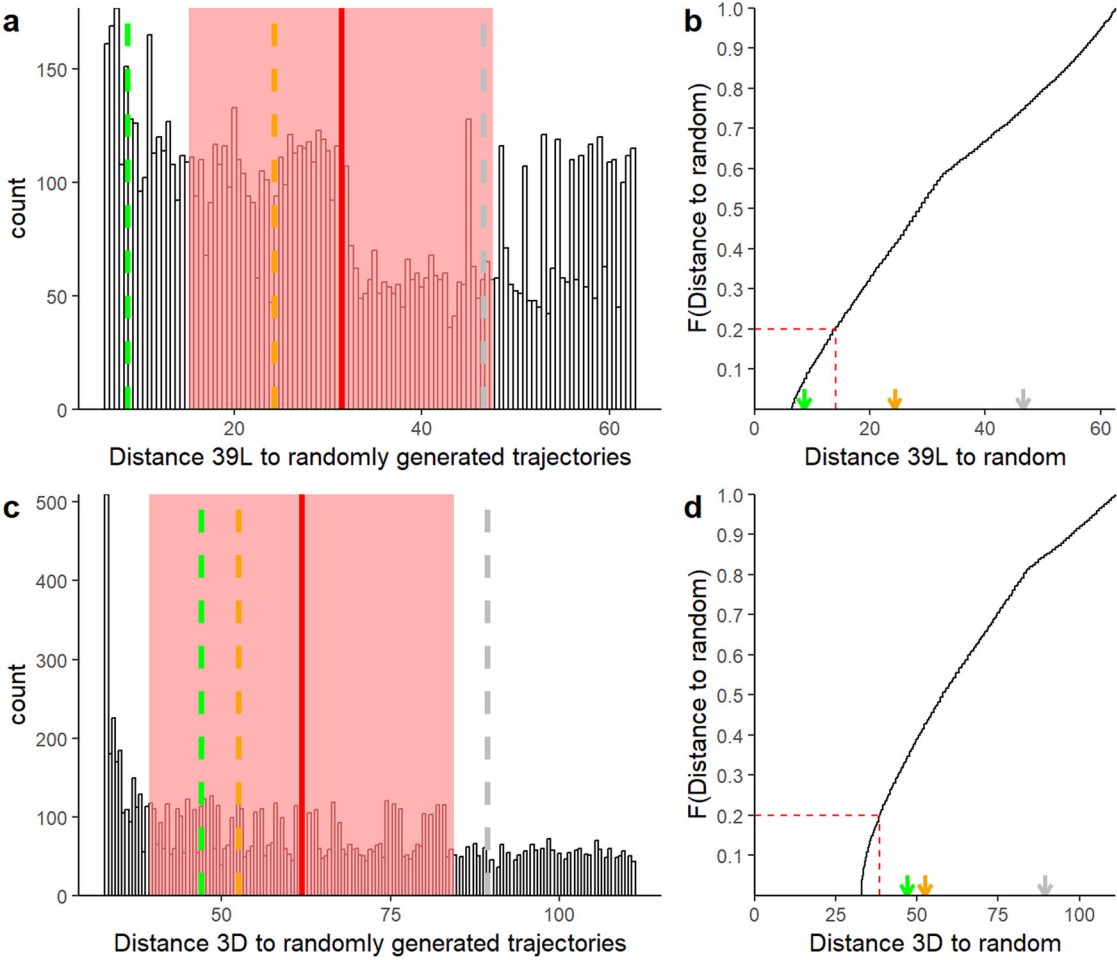

**Fig. 6 | Distribution of the distance between observed and model trajectories for individuals 39 L and 3D and the 10,000 randomly generated trajectories.** Data for individual 39 L is displayed on (**a**, **b**). Data for individual 3D is displayed on (**c**, **d**). Individuals were selected at random for illustrative purposes. Histograms (**a**, **c**), show the distribution of distances between the observed trajectory and the 10,000 randomly generated trajectories. The red line and red shaded area are the mean ± SDuniform of those distances. The dotted lines represent the mean distances

between the observed trajectories and the model trajectories (PI: green, APC: gray, RR: orange). Curves (**b**, **c**), show the empirical cumulative distribution of the distances between the fish trajectory and the 10,000 randomly generated trajectories. The dotted red line shows the 20% threshold of the distances. The arrow represents the mean values of distance between observed trajectories and model trajectories (PI: green, APC: gray, and RR: orange).

distribution for the 10,000 distances (see formula below), two fish trajectories (3 L and 29 L) would be assigned to a random trajectory instead of AC.

$$SD_{normal} = \sqrt{\frac{\sum (x_i - \mu)}{N}}$$

where N is the number of distances from random (10,000), μ is the mean distance calculated for the 10,000 distances and individual distance values are represented by $x_i$.

Finally, using a 20% threshold on the cumulative distribution function of the 10,000 distances (Fig. 6b, d) resulted in the same path categorization as that obtained using the standard deviation calculated assuming a normal distribution.

**Calculation of the fish First angle (first orientation when exiting the chamber) and Angle start-end (orientation from the first point out of the chamber to the end of the trajectory).** Each observed trajectory was made up x and y coordinate indicating the fish position and extracted every second from the time the individual excited the reward chamber. The trajectory was therefore divided into multiple segments between each coordinate points. The first direction taken by the fish was calculated as

the angle of the first straight displacement (in degrees). The first straight displacement length was the addition of segments that does not deviate by more than 10° from the previous segment. We took the coordinate of the first point out of the chamber and the coordinate from the point as the end of the first straight displacement and measure the angle of the first direction taken by the fish.

$$\text{If } y_{end} > y_1 : \theta = degree\left(arcosine\left(-\frac{x_1 - x_{end}}{\sqrt{(x_1 - x_{end})^2 + (y_1 - y_{end})^2}}\right)\right)$$

$$\text{If } y_1 > y_{end} : \theta = 360 - degree\left(arcosine\left(-\frac{x_1 - x_{end}}{\sqrt{(x_1 - x_{end})^2 + (y_1 - y_{end})^2}}\right)\right)$$

where $x_1$ and $y_1$ are the coordinates of the fish when it first exited the reward chamber and $x_{end}$ and $y_{end}$ are the coordinates of the fish at the end of the straight displacement (or at the end of the trajectory for the angle start-end).

The angle start-end was calculated using the coordinates of the first point out of the chamber and the last point before the fish turn back toward the origin.

**Comparison of model and observed angle swum**. To test whether the direction taken by the fish when first exiting the chamber was significantly different from the model prediction, a one-way bootstrap t-test was used (number of bootstrap replicates = 9999). For each grouping (PI, APC or RR), the angle of the first straight displacement of the fish trajectories was tested against the expected model angle (PI: 45°; APC: 135° ; RR: 90°). Normality of the data was tested using a Shapiro test (PI group: W = 0.93, $p$ = 0.51, APC$_{lateral}$ group W = 0.77, $p$ = 0.06, RR group W = 0.92, $p$ = 0.55). We performed the same analysis to determine if the angle start-end of the fish trajectory was different from the expected model direction (normality test: PI group: W = 0.78, $p$ = 0.02, APC$_{lateral}$ group W = 0.93, $p$ = 0.60, RR group W = 0.95, $p$ = 0.72). The expected angle start-end of RR model was 45°. When normality was not verified (PI group) we performed a signed-rank Wilcoxon test.

Only two fish were found to adopt an APC strategy after diagonal displacement. The expected angle for those two fish was 180°. However, a sample of two individuals is insufficient for statistical analysis therefore the raw values only are reported in the results.

To draw Fig. 2, we used circular statistics (package circular[43]). For each group, mean circular directions were calculated using the mean function on circular data (units = degrees). The uniformities of direction were assessed with a Rayleigh test. The length of the arrow on the figure indicate the strength of uniformity.

**Comparison of model and observed distance traveled**. A one-way $t$ test was used to test whether the first trajectory length was different from the expected model length. For each grouping (PI, APC or RR), the length of the fish trajectory was tested against the expected model length (PI: 38.6 cm; APC first vector: 38.6 cm; RR: 54.5 cm). Note that for this analysis the entire trajectory before orienting back to the origin was used, not the cropped trajectory used to group fish trajectories into different model types. Normality of the data was tested using a Shapiro test (PI: W = 0.72, $p$ = 0.003; APC: W = 0.76, $p$ = 0.047; RR: W = 0.94, $p$ = 0.67). Where normality was confirmed, a one-way bootstrap $t$ test (number of bootstrap replicates = 9999) was used. Otherwise, a signed-rank Wilcoxon test was performed. Only two fish were found to adopt an APC strategy following diagonal displacement. The expected distance for those two fish was 27.3 cm. However, a sample of two individuals is insufficient for statistical analysis therefore the raw values only are reported.

**Testing the effect of age, speed, trial number and displacement on chosen strategy**. To determine if individual age, swimming speed, trial number (first or second time in the experimental tank) and displacement direction (lateral versus diagonal) had an impact on the strategy chosen, we performed a multinomial mixed effects regression ("mblogit" function, mclogit package[46];). Strategy (PI, APC, RR or random) was added as the multinomial response variable. Fish age, swimming speed, displacement direction and trial number were added as explanatory variables. Fish identity was added as a random intercept. No overdispersion was detected (dispersion function, mclogit package[46]). A posthoc analysis was performed using "emmeans" and "contrast" functions with Holm Bonferroni adjustment (emmeans package[47]) it allowed to evaluate the effect of the explanatory variables on paired response variable (i.e., probability to follow a strategy rather than another: PI vs APC, PI vs RR, APC vs random…).

**Visual acuity of L. ocellatus**. In their review ref. 36. showed that the eye size (i.e., lens diameter) of the ray-finned fishes was significantly correlated to their visual acuity. We used the regression from ref. 36. to estimate of the visual acuity of *L. ocellatus*. The regression for fish with lens diameter below 10 mm was used (fish acuity = 2.2957 + 1.225 x lens diameter). We dissected lenses from 15 adults *L. ocellatus*, given to us by another fish lab in the department of biology, after they died from natural causes or fight with conspecific. No fish was culled to obtain their lenses.

## Reporting summary

Further information on research design is available in the Nature Portfolio Reporting Summary linked to this article.

## Data availability

All data used in the analysis is shared through Dryad open data repository (https://doi.org/10.5061/dryad.83bk3j9xr).

## Code availability

Matlab and R code used for the analyses and for the figures is shared through Dryad open data repository (https://doi.org/10.5061/dryad.83bk3j9xr).

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

## Acknowledgements

We thank Dr Olivier Bertrand for his considerable help with the statistical analysis. We thank Dr Lund for his statistical recommendations and for providing us with his expertise in circular statistics. We thank Dr Becca Goldberg for her assistance with eye dissections. We thank the Editor and three anonymous Reviewers very much for their constructive comments, which have greatly improved our manuscript. We thank Christine Soper for her help with animal husbandry, John Hogg for his help with building the experimental apparatus. This research was funded by a Human Frontier Science Program grant RGP0016/2019. C.N. was funded by a Leverhulme Trust Early Career Fellowship.

## Author contributions

A.S., C.K., and T.B. designed the study. C.N., J.E., and J.P.G. validated the experimental setup. A.S. carried out the experiment and collected the data. C.N. developed the fish tracking program. A.S. performed the statistical analyses. J.P.G. provided the individuals used in the experiment, along with information about their biology. J.E. helped with the figure design. All authors discussed the results and contributed to the final manuscript.

## Competing interests

The authors declare no competing interests.
