## [Peer Review File · Communications Biology]

Reviewers' comments:

Reviewer #1 (Remarks to the Author):

Path integration has been well studied in a number of taxa but has never been studied in fishes. Most terrestrial animals that have been studied to date tend to walk along the ground, so studying this phenomenon in fishes is unique in several ways, firstly fish don't walk and secondly, they are moving in 3D space. Showing that fish are capable of path integration certainly suggests that this capacity evolved early in vertebrates, but its existence in multiple invertebrates, particularly Hymenoptera, suggest that it is either evolved multiple times or is a basic requirement for animal navigation.

The choice of subject is clever, as these fish are very tied to their home shells, and thus like ants, are central place foragers. Likewise the methodology employed is clever and somewhat reminiscent of early studies in ants. The use of a large arena and the careful selection of relocations minimises the impacts of the surrounding walls of the arena, but is unlikely to remove it entirely. Multiple studies suggest fish can navigate based on geometric cues alone and that is borne out in the results reported here (I wonder what happens if one uses a round arena???).

While many of the fish produced random trajectories, a good number fell into categories determined a priori. The authors make good use of randomly generated data sets in order to compare their results. They find that fish used multiple methods to solve this task which also matches previous data on navigation methods in fishes. Fish, indeed many animals, often have preferred modes of navigation but can use alternative methods if conditions preclude the use of their preferred method (ie there is redundancy and likely overlap in navigation techniques).

The only thing that concerns me somewhat is that the fish were trapped in a transparent container and moved, which means they can potentially follow that movement using visual cues. This obviously effects some path integration systems but not others. I wonder what might have happened had the container been opaque?

The paper is well presented and easy to follow.

Reviewer #2 (Remarks to the Author):

Brief summary of the manuscript:

This study puts forward a behavioral assay to study path integration in a cichlid fish within laboratory conditions. Fish in a square white-walled 1.3m² tank with an overhead checkerboard pattern followed a transparent L-shaped tube from a shell territory to a mostly circular food chamber with approximately ¼ of the chamber being a flat-sided door placed in a corner of the arena. There, fish were trapped and displaced either 0.5m to the left or in the center of the tank and the shell and tube were removed. Once displaced, the door was raised, and the paths fish made were tracked. The authors modeled paths following three hypotheses: Path integration, allothetic (place) cues, and route recapitulation. The authors grouped each actual trajectory into one of these three groups based on which of the three modeled paths the cropped real data fit best. The authors then modeled random data and observed if paths in each of these three groups better fit the model of the group they were part of or of the modeled random data and further grouped the data into ultimately four groups: Path integration (n=8), allothetic (place) cues (n=6), route recapitulation (n=5), and random movements (n=18). Paths grouped in the path integration, allothetic (place) cues, and route recapitulation groups were then analyzed to compare expected directions (using two metrics) and distances travelled before fish turned back to the feeder. From these data, the authors claim that their fish exhibited paths according to all three navigational hypotheses presented as well as random paths in their experiment. Importantly, the authors highlight their claim that they have shown path integration in a fish for the first time, which to my knowledge has indeed not been reported in the literature.

Overall impression of the work:

The manuscript is overall well written, though a few comments regarding the organization of the writing can be found in the minor comments section of the review. Though the authors clearly made a great effort for both, behavioral tests and subsequent analyses, from the data displayed in manuscript, I am unfortunately unconvinced that evidence for path integration or either of the other two navigational hypotheses, allothetic cues or route recapitulation, are adequately presented in the manuscript. I think a few clarifying figures of the existing data and an additional experiment would better support the authors claims. I outline specific comments addressing these concerns below.

Major comments:

1. The displacement chamber: The geometry of the displacement chamber is not specifically described in the methods with the exception that it was transparent and 10 cm in diameter. I assume from the figures that it was circular with a single flat edge where the door is located. The angular space the door occupies from the center of the chamber should be reported. The reason this information is important is that the paths the fish make from the feeder seem to fan out in a distribution spanning 90 degrees from the door. According to the image of the chamber in Figure 1, the geometry of the chamber biases fish paths in this direction; therefore, it would not be surprising if a random distribution of fish paths from the constrains of your experimental setup closely encompasses the paths presented in your data. This is important for the next point.

2. Grouping the data based on modeled hypothesized data: The swimming trajectories displayed in Figure 3 are grouped by the four groups they were assigned to and ultimately analyzed with. However, a figure of all trajectories compiled together (I've roughly stitched the plots together and attached it) would illustrate the point I am making below. When compiled, all paths appear to create a unimodal distribution of somewhat similar looking paths present in all directions encompassing 90 degrees. Even though the logic for how your trajectories were separated into your four groups is understood, the rationale behind why the data should be segregated is lost. Perhaps a test to show that your data better fits a multimodal rather than unimodal distribution when taken together would justify separating the data into groups. By separating groups without a real justification for doing so, cherry picking may result as an unintended consequence. The 'CircMLE' package in R may be useful to determine if your data fits specific orientation distributions, at least for your angular data, which might be able to help you determine if your data better fits a unimodal or multimodal distribution. Here is the paper that describes how the package can be utilized: Fitak, R. R. and Johnsen, S. (2017) Bringing the analysis of animal orientation data full circle: model-based approaches with maximum likelihood. *Journal of Experimental Biology* 220: 3878-3882.

-The orientations of all individuals placed in an orientation plot where each trajectory orientation is a single point on the circumference of the circle (you can make this using the 'circular' package in R) would be enlightening to observe the distribution of the 37 paths you used in your analyses together, separated in groups by color on one plot. Your hypothesized orientations for the three different navigational strategies you posit are only separated by 45 degrees. Looking at examples of data from the path integrating animals you cite (which do so very well) due to error in path integration, variance in datasets often span over 45 degrees, and data from these animals using path integration in isolation rarely are as acute as reported in figure 3a. In light of this observation, if your

fish are indeed path integrating during your experiment in some cases, it is conceivable that some trajectories are grouped as random movements when they should belong to the PI group, perhaps an unintended consequence of your method of separating your data into groups.

-Since fish were grouped into each hypothesized group based on the model they were most aligned with, it seems obvious that they would be oriented in the expected modelled direction. In this case, the criterion that groups the data, which is quite strict with the addition of further pulling unideal trajectories in each hypothesis group into a random movement group, and the test of matching the criterion seem quite similar. This highlights the concern of unintended cherry-picking.

- Lines 153 and 187: For the 5 fish that you claim exhibit route recapitulation, if the fish were truly doing so, is it not surprising that none of them followed the L-shaped route back? Is it not equally plausible that these fish were also exhibiting random oriented swimming away from the shelter? One way to somewhat answer this question is to evaluate how the paths in all groups quantitatively compare to each other, including the random group? Are the lengths and straightness of paths before the turn back point different between groups?

3. For the allothetic cue group, lateral and diagonal displacement groups were analyzed in different groups while both experimental conditions were pooled for the path integration and route recapitulation groups. I believe that lateral and diagonal displacement groups should be separated for all conditions. This is due to the fact that geometries of the arenas are different between the two experimental conditions. Even though your hypothesized path for path integration is the same in both experimental conditions, animals might behave in different ways in the two conditions. This is because animals often use multiple navigational strategies in concert when relevant information is available. These strategies, while separate, are not mutually exclusive. Experiments from desert ants (Wystrach A, Mangan M, Webb B. 2015. Proc. R. Soc. B 282: 20151484. <http://dx.doi.org/10.1098/rspb.2015.1484>) and mantis shrimp (Patel RN, Cronin TW. 2020. Proc. R. Soc. B 287: 20201898. <http://dx.doi.org/10.1098/rspb.2020.1898>) offer a couple examples where multiple navigational strategies are used together. In the lateral displacement experiments, the walls of the arena offer potentially more navigationally relevant information. If you do pool them, in Figure 3 you should plot trajectories of lateral and diagonal displacements independently. If they form a unimodal distribution statistically, then some argument can be made for grouping both experiments together, though it would still be nice for the reader to be able to see this information. Sample sizes should be reported per experimental group as well in table 2. In Figure 3, I would add plots of all trajectories not separated by models but colored by the model group you've assigned them. It would be nice if you could indicate which of the two experimental conditions they belong to. Perhaps one of the conditions would have dashed trajectories.
4. A plot of trajectory distances to cutoff point of all fish per group would be useful information too. The reader can see how the distances of each trajectory differ per grouping. As stated above, distances travelled per group should be compared to one

another. If different, it may also strengthen the argument to separate trajectories per group.

- I know additional behavioral experiments are a lot of work and are a lot to ask for. Therefore, I am generally very hesitant to ask for additional experiments. However, in this case I believe an additional experiment is necessary to alleviate some of my concerns. I suggest you replicate the diagonal displacement experiment exactly with one change: The chamber should have the door open on the opposite side from where the door currently is (perhaps you can have a door on each side of the chamber). You would then be able to see if fish leave the displacement chamber in a randomly oriented fan constrained to approximate 90 degrees in the direction of the door, mirroring the results presented in your current manuscript and supporting my concerns, or swim around the chamber to the location predicted from PI, giving your interpretation of your current results much more validity and giving you stronger results to report in an updated manuscript. For clarity I've attached an illustration of this experiment.

Suggested Experiment

- Concerns and questions about analyses:
 - Line 409: Why were trajectories cropped? It seems much more appropriate to compare modeled paths to turning back points to more accurately separate trajectories into groups. By eliminating sections of the data, you will more often place trajectories into groups that they would not have otherwise belonged to.
 - Line 444: was only the PI model trajectory used or were all models used? If only the PI model was used, why was this the case? Also a typo- an extra 'and' is in the sentence here.
 - Line 512: Circular statistics methods should be used when dealing with angular data, not linear statistical methods (frequentist statistical examples include: Shapiro test for normality- instead test for Von Mises distribution if needed; Wilcoxon test- instead use Watson two-sample test, Mardia-Watson-Wheeler tests or appropriate circular tests for your specific situation). To the best of my knowledge, modelling-based approaches for circular data hasn't been explored to the extent of linear data, but a quick google search does suggest some recent modelling approaches for circular data might exist, though I have no experience with them. Just to help make potential suggestions for places to start looking into modelling based methodologies for circular data, there are some

papers in which the x and y components are split into two separate models (<https://doi.org/10.1098/rsbl.2020.0736>) or projected onto normal distributions outside the circle (<https://doi.org/10.1111/bmsp.12108> ; <https://doi.org/10.1002/env.2326>), though I can't speak for these methods as I haven't thought deeply about them. As mentioned above, the 'CircMLE' package in R can be used to determine if your data fits specific orientation distributions, which might be able to help you determine if your data better fits a unimodal or multimodal distribution. Perhaps it can be useful in other ways too?

-Some readers might be interested in seeing the goodness of fit metrics for each fish's path to the three original models and the random model to compare how much better they fit to one model than another. I'm not sure if the distances reported in Table S2 truly clarify this question. For example, in Table S2, the bolded lowest difference indicates what group each trajectory was placed. But the Means plus/minus SDs of the lowest deviating model distance often completely encompass the distances of the other models, making it seem as if there isn't a good reason to assign the trajectory to a specific model. Perhaps AICs or BICs would be useful for readers to evaluate this?

Minor Comments:

-Figure 1 and throughout text: Allothetic cues can be used to inform path integration, as you correctly state in your introduction. Your allothetic cues hypothesis for your experiment is really relying on landmarks to indicate a specific place. So perhaps another term would be useful to avoid confusion, like allothetic place cues, landmarks, or something along these lines.

-Line 34: For completeness, since you bring up PI in vertebrates and insects, terrestrial PI also shown in spiders (Seyfarth et. al, 1982) and crustaceans (in fiddler crabs; Ziel, 1998).

-Line 46: 'Patel & Cronin, 2020, Current Biology 30, 1981–1987' is more relevant to your paper than the cited work as both orientations and distances travelled are measured and displacement experiments are enacted, making it more comparable to your own work.

-Line 48: Your argument isn't exactly true here. Mantis shrimp, like fish, also have to deal with 3-dimensional navigational space. Though mantis shrimp are primarily benthic animals, they occupy complex 3D reef environments and both walk and swim while navigating. This is also often the case with many animals constrained to walking in terrestrial environments, since these environments are often not flat planes, and sometimes can be truly 3D spaces like those many insects encounter when climbing through bushes and trees.

-Line 74: Explain why or what you mean when landmark navigation becomes more error prone. Do you mean because landmarks are harder to view and may be misidentified? What do you mean by landmark navigation being slow?

-Line 91: Material in this paragraph is more appropriate in a discussion section.

-Line 122: 'An important number'- state how many fish instead.

-Line 141: The level of detail of methods presented here is probably best reserved for the methods section.

-Line 231: Since this topic is being discussed, the example in the aquatic environment should be discussed since it is directly relevant to your situation: evidence of landmark and path integration strategies working in concert in mantis shrimp (Patel RN, Cronin

TW. 2020. Proc. R. Soc. B 287: 20201898. <http://dx.doi.org/10.1098/rspb.2020.1898>).
Another desert ant example is quite a good one for this point as well (Wystrach A, Mangan M, Webb B. 2015. Proc. R. Soc. B 282: 20151484. <http://dx.doi.org/10.1098/rspb.2015.1484>).

-Line 307: With the current data presented, I think it is a stretch to claim behavioral evidence of path integration in a fish, especially with the strength of how the claim has been written here.

-Line 337: How was the arena illuminated?

-Line 546: Where in your manuscript is the visual acuity work discussed besides in the methods? Why was this work done and how does it relate to your work? To make sure the overhead checkerboard was visible? Apologies if I have missed where it is stated.

I hope you find my comments helpful. I look forward to seeing how your story progresses and I hope to see compelling evidence for path integration in a fish for the first time in the future! Best of luck!

Answer to reviewers

We would like to thank the Editor and the Reviewers very much for their constructive comments, which have greatly improved our manuscript.

Referee expertise:

Referee #1: Fish spatial cognition

Referee #2: Path integration in animals

Reviewers' comments:

Reviewer #1 (Remarks to the Author):

Path integration has been well studied in a number of taxa but has never been studied in fishes. Most terrestrial animals that have been studied to date tend to walk along the ground, so studying this phenomenon in fishes is unique in several ways, firstly fish don't walk and secondly, they are moving in 3D space. Showing that fish are capable of path integration certainly suggests that this capacity evolved early in vertebrates, but its existence in multiple invertebrates, particularly Hymenoptera, suggest that it is either evolved multiple times or is a basic requirement for animal navigation.

The choice of subject is clever, as these fish are very tied to their home shells, and thus like ants, are central place foragers. Likewise the methodology employed is clever and somewhat reminiscent of early studies in ants. The use of a large arena and the careful selection of relocations minimises the impacts of the surrounding walls of the arena, but is unlikely to remove it entirely. Multiple studies suggest fish can navigate based on geometric cues alone and that is borne out in the results reported here (I wonder what happens if one uses a round arena???).

Thank you for your comment. In this experiment, we have controlled as much as possible for the use of geometric cues by displacing the fish in two different locations in the experimental apparatus. If tested in a circular arena, we expect to observe similar results with some fish showing the ability to path integrate, other fish that might follow a route recapitulation strategy and finally some fish showing random trajectories. In a circular arena we would probably observe fewer or no fish following an allothetic cues strategy and orienting toward the original position of the shelter (indeed no geometrical information -tank edges- would be available in this case).

We agree that the fish can navigate using geometrical cues and have emphasised this point in the discussion, lines 288 to 295. We also mentioned that the use of circular arena should be favoured in further experiments, line 295.

While many of the fish produced random trajectories, a good number fell into categories determined a priori. The authors make good use of randomly generated data sets in order to compare their results. They find that fish used multiple methods to solve this task which also matches previous data on navigation methods in fishes. Fish, indeed many animals, often have preferred modes of navigation but can use

alternative methods if conditions preclude the use of their preferred method (ie there is redundancy and likely overlap in navigation techniques).

The only thing that concerns me somewhat is that the fish were trapped in a transparent container and moved, which means they can potentially follow that movement using visual cues. This obviously effects some path integration systems but not others. I wonder what might have happened had the container been opaque?

We are sorry that this important information was probably given too late in the manuscript or was not clear enough.

As soon as the door of the transparent container was closed, we wrapped the container in an opaque laminated sheet of paper. This prevented the fish from obtaining any visual information while it was moved (see below snapshots of the experimental videos for fish C3 lateral displacement. All videos of snapshots of the covered chamber are available on demand).

We chose a transparent container instead of an opaque one as during pilot trials the fish would not enter the opaque container.

To improve clarity of the methods, we have moved the information earlier in the manuscript (as soon as we mentioned that the holding chamber is transparent). Lines 394-396, 414-415 and 422-424.

Experimental apparatus at the start of the experiment

Once the fish entered the food reward chamber an opaque laminated sheet was placed around the chamber to prevent the fish from obtaining any visual cues during displacement and also while the experimenter was removing the shell and tunnel, flattening the sand, and stirring the water.

Once the fish was displaced the opaque laminated sheet was removed by the experimenter. Then the experimenter closed the curtain and opened the trap door remotely.

The paper is well presented and easy to follow. I've made multiple comments on the annotated pdf.

Thank you very much, see our answer to the comments below.

Reviewer 1 answer to comments from the annotated PDF

Line 50: also, water currents and or tides probably play havoc with distance estimation based on swimming speed.

Water currents and tides can indeed affect fish swimming speed and movements. In visually guided species, experiments by Karlsson et al. 2022 and Sibeaux et al. 2022 have shown that fish use optic flow (and not time) for distance estimation.

In these two experiments, testing teleost fish (goldfish and Picasso triggerfish) distance estimation ability, time was not used as a proxy to estimate the distance travelled (see Karlsson et al. 2022 & Sibeaux et al. 2022). Distance estimation based on swimming speed would require the fish to integrate information about their own speed and time ($\text{Distance} = \text{Time} * \text{Speed}$). In both experiments, the distance travelled and the swimming speed were significantly affected by the background pattern, indicating that the fish used the frequency of the background to estimate the distance travelled. Moreover, removing background optic flow compromised swimming speed control.

Because fish did not use time to estimate distance travel it is unlikely that these fish use their speed combined with time to estimate distance.

However, we agree that alternative mechanisms are possible and the high diversity of fish species and sensory systems will lead to a diversity of distance estimation mechanisms.

We have added these important considerations in the introduction, lines 53.

Line 63: which stands to reason given no one has shown that they can even do it.

We agree with your comment on the following manuscript sentence: "While a variety of cells that underpin spatial cognition have been found in the goldfish (e.g., speed cells, boundary vectors cells), the presence of a neural substrate for path integration in fish has not yet been identified¹³⁻

15."

Line 89: how would the fish know where the shell was, unless they can see how much they have been displaced? Perhaps you are suggesting they can use the global position within the tank (geometry) to navigate back to original position of the shell.

Yes, this is exactly what we suggest for the “allothetic cues” strategy. The fish might use allothetic information that we have not been able to conceal (i.e. the geometry of the experimental apparatus) to return to the original position of the shell.

Following your comment and another comment from reviewer 2, we decided to rename this strategy “allothetic place cues” to avoid confusion with other possible allothetic cues.

Line 96: well both sharks and agnathans predate bony fish

We acknowledge that jawless fish and cartilaginous fish are from earlier vertebrate lineage than ray-finned fish. We have changed our sentence from:

“This demonstrates that path integration is not limited to terrestrial animals, and that this ability is present in the earliest vertebrate lineage.”

to

“This demonstrates that path integration is not limited to terrestrial animals, and that this ability is present in the **early** vertebrate lineage.”

Line 100

Line 141: remove “points”

Points removed, thank you

Line 217: labrids and cyprinids are also a long way apart on the fishy phylogeny

Yes, the Picasso triggerfish (Balistidae, Order: Tetraodontiformes) and Goldfish (Cyprinidae, Order: Cypriniformes), we have added this information in the manuscript. Line 250.

Sentence: “In fact, there are clues suggesting that path integration may be widespread among teleost fish. One requirement of path integration is that the navigator can estimate distance travelled. This ability was recently demonstrated in the marine coral reef Picasso triggerfish ¹⁰, and freshwater goldfish ¹¹. These two species, **phylogenetically distant**, inhabit different environments in terms of the types of allothetic cues available, water movement, and propagation of sensory signals.”

Line 243: One possibility is that some of the fish that showed up as random do use alternative methods as priorities (Eg beackon or landmarks) but are lost in their absence

Good point, we have added this information in a later paragraph about the random trajectories. lines 331-332.

Line 276: but not in your setup considering they had a chance to explore the arena prior to testing

Our initial statement suggests that the benefits of a recapitulation strategy may be that a recently safe route is more likely to still be safe for the fish. The reviewer points out that this is not relevant in our experiment as they would already know there is no predator in the tank. We agree that the opportunity to explore the experimental apparatus overnight and before testing provides cues to the fish about the absence of a predator or barriers. However, we wanted to emphasise that in the wild this strategy could be favoured by some individuals for the reasons mentioned above and have clarified this in the text. See changes line 312.

Sentence “A few individuals followed a route recapitulation strategy, retracing their previous route. This suggests that these fish had memorised the path taken from their home shell to the food reward, and encoded it as two consecutive vectors. There are some potential benefits to this strategy **in the wild**. While a shortcut home may be faster and more energy efficient, any risks associated with the new path, such as the presence of a predator or an unpassable barrier, are unknown.”

Line 281: One thing that bugs me a little is what the chances of generating one of the expected pathways at random?

The methods do outline how the modelled trajectories were generated. This seems ok.

Indeed, in the methods, we detail how some observed trajectories were assigned to random. The fish was **first** assigned to one of the model trajectories: if the distance between the fish “observed trajectory” and the “model trajectory X” was significantly smaller than the distance to the other model trajectories Y and Z; the fish was assigned to the “model trajectory X”. **Then**, if the distance to the “model trajectory X” was smaller than the distance to “10000 random trajectories” – SD, the fish was still assigned to “model trajectory X”. If the distance to the “model trajectory X” was bigger than the distance to “10000 random trajectories” – SD, the fish was assigned to “random trajectory”.

This is equivalent to using a 20% threshold on the cumulative distribution function of the 10,000 distances to random.

If an observed trajectory follows exactly one of the model trajectories (distance to model = 0) there is no chance for this observed trajectory to be categorised as random as the distance to all random minus SD_{uniform} will be certainly above 0.

Line 338: provide width and length

1.3 m² means that the experimental apparatus was 1.3 m wide and 1.3 m long. We have specified those values along with the heights in the brackets line 387.

Line 344: transparent holding chamber highlighted

We have added in the following sentence that the transparent chamber will be covered by an opaque laminated sheet for displacement. See lines 394-396.

Line 365: because it is transparent they can actually see that they are being moved

As mentioned above we had an opaque laminated sheet over the chamber before displacement so the fish could not pick up any visual information about its displacement. We added the information lines 394-396.

Line 367: clever

Thank you!

Reviewer #2 (Remarks to the Author):

Brief summary of the manuscript:

This study puts forward a behavioral assay to study path integration in a cichlid fish within laboratory conditions. Fish in a square white-walled 1.3m² tank with an overhead checkerboard pattern followed a transparent L-shaped tube from a shell territory to a mostly circular food chamber with approximately ¼ of the chamber being a flat-sided door placed in a corner of the arena. There, fish were trapped and displaced either 0.5m to the left or in the center of the tank and the shell and tube were removed. Once displaced, the door was raised, and the paths fish made were tracked. The authors modeled paths following three hypotheses: Path integration, allothetic (place) cues, and route recapitulation. The authors grouped each actual trajectory into one of these three groups based on which of the three modeled paths the cropped real data fit best. The authors then modeled random data and observed if paths in each of these three groups better fit the model of the group they were part of or of the modeled random data and further grouped the data into ultimately four groups: Path integration (n=8), allothetic (place) cues (n=6), route recapitulation (n=5), and random movements (n=18). Paths grouped in the path integration, allothetic (place) cues, and route recapitulation groups were then analyzed to compare expected directions (using two metrics) and distances travelled before fish turned back to the feeder. From these data, the authors claim that their fish exhibited paths according to all three navigational hypotheses presented as well as random paths in their experiment. Importantly, the authors highlight their claim that they have shown path integration in a fish for the first time, which to my knowledge has indeed not been reported in the literature.

Overall impression of the work:

The manuscript is overall well written, though a few comments regarding the organization of the writing can be found in the minor comments section of the review. Though the authors clearly made a great effort for both, behavioral tests and subsequent analyses, from the data displayed in manuscript, I am unfortunately unconvinced that evidence for path integration or either of the other two navigational hypotheses, allothetic cues or route recapitulation, are adequately presented in the manuscript. I think a few clarifying figures of the existing data and an additional experiment would better support the authors claims. I outline specific comments addressing these concerns below.

We would like to thank reviewer 2 very much for their detailed reading of our manuscript and for their very thoughtful comments. In light of these comments, we have made some changes to our manuscript and performed additional analyses. Together, these amendments have greatly improved our manuscript and the strength of the analyses.

Major comments:

1. The displacement chamber: The geometry of the displacement chamber is not specifically described in the methods with the exception that it was transparent and 10 cm in diameter. I assume from the figures that it was circular with a single flat edge where the door is located. The angular space the door occupies from the

center of the chamber should be reported. The reason this information is important is that the paths the fish make from the feeder seem to fan out in a distribution spanning 90 degrees from the door. According to the image of the chamber in Figure 1, the geometry of the chamber biases fish paths in this direction; therefore, it would not be surprising if a random distribution of fish paths from the constraints of your experimental setup closely encompasses the paths presented in your data. This is important for the next point.

Working with fish required adjustment to the already existing paradigms performed by our peers to test path integration abilities in ants, mantis shrimps or rodents. Fish have very sensitive and specialised sensory systems (such as their lateral line) that give them access to a high number of cues unavailable to humans or other species (such as their ability to detect fine differences in hydrodynamic movement). We needed to control for multiple extra cues, available to our model species, in this experiment. In our experimental design, it was essential to trap the fish (and cover the trap) in order to remove experimental landmarks, mix the water and the sand and displace the fish. Our experiment was designed to be the most appropriate for the species that we studied and to be able to observe the most natural and relevant behaviours. This adapted paradigm, including a trap chamber, has allowed us to test their ability to perform path integration strategy.

The chamber was 9.5 cm (diameter) x 19.5 (height) transparent cylinder, with a 6.5 cm diameter circular opening. The rectangular door was attached to the inner walls of the chamber. We included a scaled illustration in Figure 1, line 107.

From the centre of the chamber, the fish had indeed a 90-degree angle of freedom of movement. However, the first recorded position (time=0) was on the outer edge of the chamber, from where the fish had 180-degree angle freedom of movement (see individual trajectory pictures in the supplementary material). Some fish, such as 3L, 29L, 10D or 26D took directions perpendicular to the door opening (see 26D example below and supplementary figures S1a and S1b for all other trajectories) indicating that fish were able to choose an angle from 0 to 180 degrees.

Crucially, our data do not follow either a unimodal or a uniform distribution but follow a multimodal distribution (M5A=heterogeneous bimodal, see extra circular analyses performed to answer major concern #2 and supplementary material Tables S2 a and b). Our experimental apparatus did not prevent the fish from following a range of statistically different orientations when exiting the reward chamber (Watson-Wheeler circular test for homogeneity of means on PI, RR, ACP and random circular data: $W = 24.77$, $df = 8$, $p\text{-value} = 0.002$).

Finally, we measured the fish speed when exiting the reward chamber. In the two seconds immediately after the fish exited the chamber, their speed was relatively slow (average = $5.3 \text{ cm}\cdot\text{sec}^{-1}$ _ without outlier trial 15L _ see raw data on dryad and supplementary material S1a & S1b for fish trajectories). Thus, in a second, they were approximately swimming a distance that approximated their standard body length. We believe this slower swimming speed gave the fish enough time to adjust their swimming direction to account for the narrowing of the door opening.

Together, we can conclude that the fish were able to follow a range of trajectories and did not follow a unimodal direction averaging 90° when exiting the chamber.

2. Grouping the data based on modeled hypothesized data: The swimming trajectories displayed in Figure 3 are grouped by the four groups they were assigned to and ultimately analyzed with. However, a figure of all trajectories compiled together (I've roughly stitched the plots together and included it in the attached PDF) would illustrate the point I am making below. When compiled, all paths appear to create a unimodal distribution of somewhat similar looking paths present in all directions encompassing 90 degrees. Even though the logic for how your trajectories were separated into your four groups is understood, the rationale behind why the data should be segregated is lost. Perhaps a test to show that your data better fits a multimodal rather than unimodal distribution when taken together would justify separating the data into groups. By separating groups without a real justification for doing so, cherry picking may result as an unintended consequence. The 'CircMLE' package in R may be useful to determine if your data fits specific orientation distributions, at least for your angular data, which might be able to help you determine if your data better fits a unimodal or multimodal distribution. Here is the paper that describes how the package can be utilized: Fitak, R. R. and Johnsen, S. (2017) Bringing the analysis of animal orientation data full circle: model-based approaches with maximum likelihood. *Journal of Experimental Biology* 220: 3878-3882.

We understand your concern as we also spent a lot of time evaluating the best methods to analyse these data. Both the authors of the circular package **Dr. Ulric J Lund** and a Specialist of Monte Carlos analysis **Dr Olivier Bertrand**, were contacted to make sure that we were not missing anything in the analysis.

- 1) Dr Lund recommended that we not use the circular package for our data, given that our data is restricted to 0-180°. Dr Lund recommended that we compute the circular mean direction as a descriptive statistic, but beyond that, he recommended that we perform ordinary statistics for our data.**

Indeed, in our case, we do not have the problem of 355° being closer to 5° than 330°. In our case 180° is the further angle from 0° and angles are linearly further apart.

We agreed with Dr Lund and performed ordinary statistics that are robust and powerful to analyse our data.

We also used circular statistics to compute the circular means for each group and measure the strength of the uniformity with a Rayleigh test (see Figure 2). We also decided to analyse our data with the package that you mentioned (see 3rd point of this discussion) as a way to support our grouping method.

- 2) Moreover, we tested grouping options using MC3 analysis with the help of Dr Bertrand.**

The classification on frechet and DTW similarity measures was used on the cichlid trajectories. From this analysis, the optimal number of path clusters was 4 and this was significantly different from 1 (i.e. no cluster). As you can see on the figure below.

The classification was greatly influenced by the endpoint of the cichlid. Since both metrics compare every point in trajectory with another trajectory, the distance increases greatly when the two endpoints are far apart (see below).

To minimize this impact, the trajectories were truncated to the smallest distance travelled

by the fish. After repeating the test, we still obtained 4 clusters, separated from left to right (also probably driven by the endpoint).

For this reason, Dr Bertrand recommended we not use this method to classify our trajectories but agreed that multiple clusters are better than none.

3) We tested the package that you suggested (circ_mle) to analyse our data.

We would like to thank you for hinting at this paper as it was a circular analysis method and a package that we were not aware of before reading your comments. We feel that it has helped clarify and improve the global analysis.

This package appears to provide more accurate results and a higher level of detail than the simple Rayleigh test from the circular package (Fitak & Johnsen, 2017). **Before discussing our circular analysis, we want to note that our data ranges from 0 to 180°, therefore, circular statistics on these data should be interpreted with care.**

After reading the paper you suggested, we ran a circ_mle test on our data. We preliminarily transformed to circular data using the information provided in the paper. We decided to change modulo to pi compared to 2pi used on the paper as our data can only spread from 0 to Pi.

```
dc<-circular(Angle.first.straight.path, units = "radian", modulo = "pi").
```

We obtained the following output from circ_mle(dc) and the plot function:

	M5A	M2A	M2C	M2B	M3B	M4B	M5B	M4A	M3A	M1
Params	4	2	3	2	3	4	5	3	2	0
$\phi 1$	0.61	1.87	2.11	2.13	5.26	5.43	6.28	5.11	2.27	NA
$\kappa 1$	4.63	1.98	3.77	4.18	0.00	0.00	5.95	2.00	1.82	0.00
λ	0.25	1.00	0.75	0.50	0.50	0.35	0.40	0.25	0.50	1.00
$\phi 2$	2.19	NA	NA	NA	8.41	8.57	2.23	8.25	5.41	NA
$\kappa 2$	4.63	0.00	0.00	0.00	4.54	5.00	5.06	2.00	1.82	0.00
Likelihood	42.63	47.13	50.24	52.32	52.35	52.06	51.18	55.08	65.21	68.00
Convergence	0	0	0	0	0	0	0	0	0	0
AIC	93.26	98.26	106.48	108.64	110.70	112.12	112.36	116.16	134.41	136.00
AICc	94.51	98.61	107.21	109.00	111.42	113.37	114.30	116.88	134.77	136.00
BIC	99.70	101.48	111.31	111.87	115.53	118.57	120.42	120.99	137.64	136.00
Δ AIC	0.00	5.00	13.22	15.39	17.44	18.87	19.11	22.90	41.16	42.75

$\Delta AICc$	0.00	4.10	12.70	14.49	16.92	18.87	19.79	22.38	40.26	41.50
ΔBIC	0.00	1.78	11.61	12.17	15.83	18.87	20.72	21.29	37.94	36.30
Relative likelihoods	1.00	0.08	0.00	0.00	0.00	0.00	0.00	0.00	0.00	0.00
AIC weights	0.92	0.08	0.00	0.00	0.00	0.00	0.00	0.00	0.00	0.00
Evidence Ratio	NA	1.22E+01	7.43E+02	2.19E+03	6.12E+03	1.25E+04	1.41E+04	9.38E+04	8.65E+08	1.92E+09

Figure 1: Raw plot from the circ_mle function. The original parameters from the plot have been left intact meaning that the angular values are in radians units and the values are plotted counter clockwise.

According to this test, and following the model selection procedure based on AIC, the best-selected model for our data is model M5A = homogeneous bimodal ($\Delta AIC=0$, see attached figure, and supplementary material).

The same test performed on the angle from the start of the trajectory to the end of the trajectory led to the selected model M5B=Bimodal ($\Delta AIC=0$, code output table and figure, available on the supplementary material)

We also ran this test with modulo= 2π , to evaluate if it led to any differences. In this case, the default model selection procedure (AIC) selected model M3A= Homogeneous symmetric bimodal, as the best ($\Delta AIC=0$, code output table and figure, available on demand).

A further circular analysis performed to answer your concern #6, lead us to run a non-parametric Watson-Wheeler circular test for homogeneity of means on two or more samples of circular data. We used the Watson-Wheeler circular test for our four groups (PI,APC, RR and Random). The output of this test ($W = 24.77$, $df = 8$, $p\text{-value} = 0.002$) indicates that our groups were significantly not homogeneous.

The results from the discussion with Dr Lund, Dr Bertrand and the latest “circ-mle” analysis performed, led us to conclude that our data do not follow a unidirectional or unimodal distribution (circular analysis) and that grouping can be applied and is significantly better for our data (MC3 analysis). This additional circular data analysis adds strength to our original conclusions. We believe our original grouping method (smallest distance to one of the model trajectories and comparison to 10,000 random trajectories) is relevant and leads to the most

conservative results (ensuring that some of the fish tested are certainly following the model trajectory defined instead of showing random movements).

We have decided to add the circular analyses that you recommended in the methods (lines 455 to 466) and results (lines 150-159) with all tables and figures given in supplementary material (Tables S2a and S2a). This will provide the reader with information about the no-unimodal and no-unidirectional distribution of the data to support our grouping methods.

-The orientations of all individuals placed in an orientation plot where each trajectory orientation is a single point on the circumference of the circle (you can make this using the 'circular' package in R) would be enlightening to observe the distribution of the 37 paths you used in your analyses together, separated in groups by color on one plot.

We have re-drawn Figure 2 and included a single point on the circumference of the circle for each individual orientation. Line 172

Your hypothesized orientations for the three different navigational strategies you posit are only separated by 45 degrees. Looking at examples of data from the path integrating animals you cite (which do so very well) due to error in path integration, variance in datasets often span over 45 degrees, and data from these animals using path integration in isolation rarely are as acute as reported in figure 3a. In light of this observation, if your fish are indeed path integrating during your experiment in some cases, it is conceivable that some trajectories are grouped as random movements when they should belong to the PI group, perhaps an unintended consequence of your method of separating your data into groups.

You are right, the path integration error can lead to variance spanning over 45 degrees. However, the path integration studies cited allow 360-degree freedom of movement for the animal to return to the original position of its shelter. Having a restricted movement direction in our experiment and a short distance between the chamber and the expected position of the shell/ relative to the animal size we think that this should be reduced in our case. However, we agree that further experiments would be necessary to quantify the 'normal' degree of error in path integration in *Lamprologus ocellatus*.

This study aimed to provide evidence that *Lamprologus ocellatus* can path integrate. We agree that it is possible that some individuals performing path integration could have been misclassified in the random trajectory group as our analysis was very conservative. However, this more conservative approach allows us to be certain that at least some individuals were following a path integration trajectory. **We have added a full paragraph in the discussion to consider this possibility. Lines 320-328.**

Classifying our trajectories into groups was essential knowing the inter and intra-variability observed in fish navigation (see discussion for references) and it represents one of the strengths in our analysis (with post-validation for this method via the model-based approach circular analysis showing that the data are not unimodal). Furthermore, we tested fish of different ages and populations (see result section: *Effect of age, swimming speed, test trial number and displacement on the chosen strategy*) and obtained statistical significance that could explain the diversity of behaviour observed.

-Since fish were grouped into each hypothesized group based on the model they were most aligned with, it seems obvious that they would be oriented in the expected modelled direction. In this case, the criterion that groups the data, which is quite strict with the addition of further pulling unideal trajectories in each hypothesis group into a random movement group, and the test of matching the criterion seem quite similar. This highlights the concern of unintended cherry-picking.

The analyses followed two steps that we believe are crucial to obtain appropriate results and an integrated picture of the navigational behaviour of our model species. **First**, we determined if the observed trajectory was significantly closer to one of the model trajectories. **Observed trajectories were grouped by distance to the model trajectories.** Once the trajectory has been categorised within a certain model trajectory we performed an analysis of this distance against 10,000 randomly generated trajectories, allowing us to reject trajectories that could have been following random movement. However, this step did not allow us to determine how well the observed trajectories matched the model expectation. Therefore, **secondly**, we checked if the grouped trajectories fit the expectations of the model trajectory. **We tested angles and lengths.** This allowed us to determine how close the groups matched our hypotheses. If the original grouping made in the first instance was wrong this step would have allowed us to detect it (with $p < 0.05$, for all groups).

This step also allowed us to observe that the fish group categorised under route recapitulation followed the expected model orientation for the first angle and not for the angle start-end. This could be explained by the fact that the fish only retraced the first part of the L-shaped tunnel before orienting back toward the reward chamber. Line 212-213.

- Lines 153 and 187: For the 5 fish that you claim exhibit route recapitulation, if the fish were truly doing so, is it not surprising that none of them followed the L-shaped route back? Is it not equally plausible that these fish were also exhibiting random oriented swimming away from the shelter? One way to somewhat answer this question is to evaluate how the paths in all groups quantitatively compare to each other, including the random group? Are the lengths and straightness of paths before the turn back point different between groups?

We measured the lengths and straightness index (Distance / Length, Batschelet, 1981) for each trajectory (reported in the table below) and evaluated if the lengths and straightness of paths before the turn-back point were different between groups.

Table S5: Average \pm SD path length and straightness for the fish grouped under Path integration (PI), Allothetic place cues (APC), Route recapitulation (RR) and Random trajectories groups.

	PI	APC-Diagonal	APC-lateral	RR	Random
path length	41.81 \pm 41.38	145.91 \pm 185.63	94.24 \pm 75.04	25.59 \pm 8.65	79.14 \pm 56.42
path straightness	0.89 \pm 0.08	0.70 \pm 0.38	0.72 \pm 0.28	0.92 \pm 0.06	0.85 \pm 0.10

There was a significant difference in path length between the groups ($\chi^2 = 10.05$, $df=4$, $P=0.04$) but this difference was only significant for the RR fish compared to the Random fish (posthoc pairwise Wilcoxon test with Bonferroni holm adjustment $P=0.049$). There were no significant differences in path straightness between groups ($\chi^2 = 2.13$, $df=4$, $P=0.71$).

We found a statistical difference in path length between the Random and RR groups which is consistent with the route recapitulation group being different from a random group (see further discussion in major comments #4 answer). Given this, we have kept the route recapitulation group as an entity and we have not merged it with the random group.

We added the statistical test in the result section (lines 214 to 217) and the table in the supplementary material (Table S5).

3. For the allothetic cue group, lateral and diagonal displacement groups were analyzed in different groups while both experimental conditions were pooled for the path integration and route recapitulation groups. I believe that lateral and diagonal displacement groups should be separated for all conditions. This is due to the fact that geometries of the arenas are different between the two experimental conditions. Even though your hypothesized path for path integration is the same in both experimental conditions, animals might behave in different ways in the two conditions. This is because animals often use multiple navigational strategies in concert when relevant information is available. These strategies, while separate, are not mutually exclusive. Experiments from desert ants (Wystrach A, Mangan M, Webb B. 2015. Proc. R. Soc. B 282: 20151484. <http://dx.doi.org/10.1098/rspb.2015.1484>) and mantis shrimp (Patel RN, Cronin TW. 2020. Proc. R. Soc. B 287: 20201898. <http://dx.doi.org/10.1098/rspb.2020.1898>) offer a couple examples where multiple navigational strategies are used together. In the lateral displacement experiments, the walls of the arena offer potentially more navigationally relevant information. If you do pool them, in Figure 3 you should plot trajectories of lateral and diagonal displacements independently. If they form a unimodal distribution statistically, then some argument can be made for grouping both experiments together, though it would still be nice for the reader to be able to see this information. Sample sizes should be reported per experimental group as well in table 2. In Figure 3, I would add plots of all trajectories not separated by models but colored by the model group you've assigned them. It would be nice if you could indicate which of the two experimental conditions they belong to. Perhaps one of the conditions would have dashed trajectories.

We grouped the trajectories for path integration and route recapitulation after both lateral and diagonal displacement because our predictions for model trajectories for path integration and route recapitulation were the same independently of the displacement. We

agree with you that following the different displacements, individuals might behave differently because of the external information available and we have therefore done another analysis as you suggested. We used the same previously recommended R package. We tested if individual groups under the path integration group and the route recapitulation group followed a unimodal or bimodal distribution.

The best model that described the path integration trajectories was M2A= Unimodal distribution ($\Delta AIC=0$, see table and figure below).

	M2A	M2C	M2B	M4A	M3B	M5B	M5A	M4B	M3A	M1
Params	2	3	2	3	3	5	4	4	2	0
$\phi 1$	2.13	2.05	2.16	5.37	2.17	2.53	5.78	5.16	2.13	NA
$\kappa 1$	4.67	3.86	5.42	3.08	5.38	3.02	4.00	5.00	4.64	0.00
λ	1.00	0.75	0.50	0.25	0.50	0.55	0.25	0.25	0.50	1.00
$\phi 2$	NA	NA	NA	8.52	5.31	1.74	2.30	8.30	5.28	NA
$\kappa 2$	0.00	0.00	0.00	3.08	0.00	2.04	4.00	2.66	4.64	0.00
Likelihood	5.73	7.29	8.79	8.49	8.79	7.17	8.48	8.83	11.27	14.70
Convergence	0	0	0	0	0	0	0	0	0	0
AIC	15.45	20.58	21.58	22.99	23.59	24.35	24.96	25.66	26.54	29.41
AICc	17.85	26.58	23.98	28.99	29.59	54.35	38.30	38.99	28.94	29.41
BIC	15.61	20.82	21.74	23.22	23.82	24.75	25.28	25.98	26.70	29.41
ΔAIC	0.00	5.13	6.13	7.53	8.13	8.90	9.51	10.21	11.08	13.95
$\Delta AICc$	0.00	8.73	6.13	11.13	11.73	36.50	20.44	21.14	11.08	11.55
ΔBIC	0.00	5.21	6.13	7.61	8.21	9.13	9.67	10.36	11.08	13.80
Relative likelihoods	1.00	0.08	0.05	0.02	0.02	0.01	0.01	0.01	0.00	0.00
AIC weights	0.84	0.06	0.04	0.02	0.01	0.01	0.01	0.01	0.00	0.00
Evidence Ratio	NA	12.98	21.39	43.24	58.34	85.44	116.21	164.42	255.11	1071.48

Due to this result, we have decided to keep PI trajectory data pooled together independently of the displacement followed by the fish, and we did similarly for the RR group.

- We have added the sample size for individuals following lateral and diagonal displacement in Table 1.
- We have added the sample size per experimental group in Table 2.
- We changed Figure 3, all trajectories are now presented on the same plot and coloured by the model group. The two experimental conditions (lateral and diagonal displacement) can be differentiated by the line type (plane versus dashed).

4. A plot of trajectory distances to cutoff point of all fish per group would be useful information too. The reader can see how the distances of each trajectory differ per grouping. As stated above, distances travelled per group should be compared to one another. If different, it may also strengthen the argument to separate trajectories per group.

Thank you for this idea, we added a box plot of the trajectory's distances for each group (and individual data points, see below = new Figure 4 in the manuscript, line 223)

Figure 4: Swimming distance for the fish following PI, APC, RR or random orientation. Individual distances of the fish full trajectory (before the fish orientates back to the reward chamber) are represented by the grey dots. Black dots represent the average distance for each group.

As reported earlier and added in the supplementary material only RR and Random trajectories show a small but significant statistical difference in their distance.

For the analysis of groups' angles and distances observed versus expectation, we have concluded that the individuals following route recapitulation strategies seem to follow the first part of the route recapitulation trajectory only and we suggested in the discussion that individuals might have encoded it as two consecutive vectors. On the contrary, individuals following random trajectories showed an extended exploration of the experimental apparatus before returning to the chamber.

The small number of data points in some groups (e.g. APC) did not allow us to observe any other significance.

5. I know additional behavioral experiments are a lot of work and are a lot to ask for. Therefore, I am generally very hesitant to ask for additional experiments. However, in this case I believe an additional experiment is necessary to alleviate some of my concerns. I suggest you replicate the diagonal displacement experiment exactly with one change: The chamber should have the door open on the opposite side from where the door currently is (perhaps you can have a door on each side of the chamber). You would then be able to see if fish leave the displacement chamber in a randomly oriented fan constrained to approximate 90 degrees in the direction of the door, mirroring the results presented in your current manuscript and supporting my concerns, or swim around the chamber to the location predicted from PI, giving your interpretation of your current results much more validity and giving you stronger results to report in an updated manuscript. For clarity I've attached an illustration of

this experiment viewable in the attached PDF.
Suggested Experiment

We think that this extra experiment is not necessary for the following reasons:

- 1) The major reason is the non-unimodal distribution of the fish trajectories. From this non-uniform dataset, an APC, a RR and a PI pattern could be clearly distinguished from the random using appropriate stats.
- 2) Once the door is open the fish had 180-degree freedom of movement, allowing them to follow multiple navigational cues. Our experimental paradigm rigorously considered all different possible scenarios in fish navigational strategies and tested them against random movement. We were able to clearly separate the three different scenarios and used a very conservative cut-off to group our fish (see later point 4).
- 3) The displacement has been made in two different positions. This gave the fish access to a different view once they had exited the chamber. The fish did show PI trajectories (and the other model trajectories) in these two displacements. We think that this is a strength of our experiment and analyses. In most path integration displacement experiments the individuals were displaced to a single location.
- 4) Moreover, thanks to the two possible displacements, the inverse of flipping the door around 180 to check for PI trajectories was essentially done with respect to the allothetic trajectories (APC, figure 1d). The fish had to follow different trajectories to perform APC strategies after the lateral or diagonal displacements (the trajectory they had to follow for APC strategy after diagonal displacement is the mirrored trajectory that the one you suggest here for an individual that would follow PI if the door was opened on the back of the chamber). Our result showed that fish followed the APC strategy after the two displacements and therefore indicates that fish would be able to perform PI trajectory in another configuration.
- 5) Our statistical analysis was purposely conservative to ensure that the fish that followed PI trajectories, or the other model trajectories, were not performing random behaviour. We hope that the extra analyses that we detailed above have assured you and that you agree with our interpretations.
- 6) Ethical consideration: We followed the 3Rs guidelines in all our experiments and have used the minimum adequate number of individuals allowing us to answer our research question. Running further experimental trials would require using an

important number of extra individuals (as detailed in the manuscript the participation rate to the experiment was low). We think that this would be against the 3Rs recommendations.

For the reasons listed above, we believe that extra experiments would not actually add more information to our current results. From a practical point of view, changes to our facilities since the experiment was performed in 2021 also mean we would be unable to replicate the experiment under the exactly same conditions.

6. Concerns and questions about analyses:

-Line 409: Why were trajectories cropped? It seems much more appropriate to compare modeled paths to turning back points to more accurately separate trajectories into groups. By eliminating sections of the data, you will more often place trajectories into groups that they would not have otherwise belonged to.

We needed to crop the observed fish trajectory to the size of the model trajectory OR the model trajectory to the size of the observed fish trajectory (in case the fish trajectory was shorter) because we needed to interpolate 1000 points along both trajectories to obtain information about the distance between both trajectories. If full trajectories were kept, the distance between points would be different in the observed and the model trajectories and between observed trajectories leading to bias and non-comparable results.

-Line 444: was only the PI model trajectory used or were all models used? If only the PI model was used, why was this the case? Also a typo- an extra 'and' is in the sentence here.

The 10,000 random trajectories were randomly generated as straight lines (such as PI and ACP trajectories) with 10,000 randomly generated angles (spanning from 0 to 180 degrees). The random trajectories were compared to the three different model trajectories.

Thank you for pointing out the typo

-Line 512: Circular statistics methods should be used when dealing with angular data, not linear statistical methods (frequentist statistical examples include: Shapiro test for normality- instead test for Von Mises distribution if needed; Wilcoxon test- instead use Watson two-sample test, Mardia-Watson-Wheeler tests or appropriate circular tests for your specific situation). To the best of my knowledge, modelling-based approaches for circular data hasn't been explored to the extent of linear data, but a quick google search does suggest some recent modelling approaches for circular data might exist, though I have no experience with them. Just to help make potential suggestions for places to start looking into modelling based methodologies for circular data, there are some papers in which the x and y components are split into two separate models (<https://doi.org/10.1098/rsbl.2020.0736>) or projected onto normal distributions outside the circle (<https://doi.org/10.1111/bmsp.12108> ; <https://doi.org/10.1002/env.2326>), though I can't speak for these methods as I haven't thought deeply about them. As mentioned

above, the 'CircMLE' package in R can be used to determine if your data fits specific orientation distributions, which might be able to help you determine if your data better fits a unimodal or multimodal distribution. Perhaps it can be useful in other ways too?

As mentioned earlier we have been in touch with Dr Ulric J Lund from the start of our analyses. Dr Lund was contacted because he is a circular statistic specialist and one of the authors of the circular package on R. Dr Lund recommended not to use circular statistics to analyse our data. We would be happy to provide the full email exchange on demand. Based on your recommendation, we have performed some extra analyses with the circ-mle package. These tests confirmed that our data are not following unimodal or unidirectional directions and that it makes sense to group them under multiple navigation strategies. We have added this information in the results and supplementary material, Table S2. However, for the analysis presented in Table 2 (see below and the associated methods line 562), we decided to keep following Dr Lund advice because our angle analysis spans from 0 to 180 degrees and therefore does not span over a full circle. In this case, regular inferential statistics are more powerful and using circular statistics did not change the results output and significance (see below).

Table 2

Fish group	Travel metric	t	df	V	p	mean	95% CI	Model expectation
PI (n=8)	First straight angle	1.03	7	-	0.356	50.44	[40.51 ; 59.64]	45 degrees
	Angle start-end	-	-	32	0.055	63.9	[43.92 ; 83.87]	45 degrees
	Distance travelled	-	-	14	0.641	41.81	[13.14 ; 70.49]	38.6 cm
APC Lateral (n=4)	First straight angle	0.14	3	-	0.622	136.94	[112.91 ; 160.88]	135 degrees
	Angle start-end	-1.23	3	-	0.240	124.4	[108.21 ; 136.93]	135 degrees
	Distance travelled	-	-	10	0.125	94.24	[20.69 ; 167.78]	38.6 cm
APC Diagonal (n=2)	First straight angle	two observations only: 135.00 and 161.56						180 degrees
	Angle start-end	two observations only: 44.38 and 172.69						180 degrees
	Distance travelled	two observations only: 14.65 and 277.16						27.3 cm
RR (n=5)	First straight angle	-0.64	4	-	0.498	82.35	[62.21 ; 101.92]	90 degrees
	Angle start-end	5.23	4	-	0.005	80.05	[69.76 ; 92.33]	45 degrees
	Distance travelled	-7.48	4	-	0.023	25.62	[19.17 ; 32.02]	54.5 cm

We also carefully read the different papers that you mentioned. "Spatial orientation of social caterpillars is influenced by polarized light", by Uemura et al. 2021 was particularly inspiring. We also used Landers et al. 2018 "Circular data in biology: advice for effectively implementing statistical procedures" and Landers et al. 2021 "Comparing two circular distributions: advice for effective implementation of statistical procedures in biology" to obtain more information about all possible tests (detailed in the appendix of Landers et al. 2021).

We have performed the non-parametric Watson-Wheeler circular test for homogeneity of means on two or more samples of circular data. The output of this test ($W = 24.77$, $df = 8$, p -value = 0.002) indicates that our groups were significantly not homogeneous. Once again, this result supports the analysis that we performed. We have also run Watson two-sample test for homogeneity between our navigation strategy groups and randomly generated von Mises distribution data ($r_{vonmises}$, $n=n$ from the group tested, $\mu =$ expected direction of the group). The output of the Watson test was "Do Not Reject Null Hypothesis" (i.e. the mean orientation of our group was not significantly different to the expected mean) for all groups tested except the Angle Start-End for the RR group. The circular statistics test did not bring us any differences in terms of statistical significance, therefore, for the reasons mentioned above we have kept our table line 200.

-Some readers might be interested in seeing the goodness of fit metrics for each fish's path to the three original models and the random model to compare how much better they fit to one model than another. I'm not sure if the distances reported in Table S2 truly clarify this question. For example, in Table S2, the bolded lowest difference indicates what group each trajectory was placed. But the Means plus/minus SDs of the lowest deviating model distance often completely encompass the distances of the other models, making it seem as if there isn't a good reason to assign the trajectory to a specific model. Perhaps AICs or BICs would be useful for readers to evaluate this?

We have added AIC and BIC values to Table S3 (which was table S2 in the original submission, see supplementary material).

We choose to present SD error bars which are descriptive error bars (Cumming et al. 2007- Errors bars in experimental biology). These error bars indicate the average difference between the data point and their mean, and should not be used to predict statistical significance (Cumming et al. 2007). We could use inferential error bars such as SE or CI if you want us to make the change (SE are actually small and SE around each mean do not overlaps). However, we choose to have SD presented for all means here as random-SD_{uniform} for the criteria to categorise a trajectory into one of the model trajectories or to random movement trajectory.

Minor Comments:

-Figure 1 and throughout text: Allothetic cues can be used to inform path integration, as you correctly state in your introduction. Your allothetic cues hypothesis for your experiment is really relying on landmarks to indicate a specific place. So perhaps another term would be useful to avoid confusion, like allothetic place cues, landmarks, or something along these lines.

We agree with you and we have decided to rename the allothetic cues “allothetic place cues” which is more precise and prevents confusion with other allothetic cues a fish could use to navigate (e.g. magnetic field, polarisation pattern...)

-Line 34: For completeness, since you bring up PI in vertebrates and insects, terrestrial PI also shown in spiders (Seyfarth et. al, 1982) and crustaceans (in fiddler crabs; Ziel, 1998).

Thank you for these extra references, we have read them and added them to the text. Line 36.

-Line 46: ‘Patel & Cronin, 2020, Current Biology 30, 1981–1987’ is more relevant to your paper than the cited work as both orientations and distances travelled are measured and displacement experiments are enacted, making it more comparable to your own work.

Thank you very much, this is the work we wanted to cite, there has been a mistake made with Mendeley, our citation add-ins. Now corrected line 48.

-Line 48: Your argument isn’t exactly true here. Mantis shrimp, like fish, also have to deal with 3-dimensional navigational space. Though mantis shrimp are primarily benthic animals, they occupy complex 3D reef environments and both walk and swim while navigating. This is also often the case with many animals constrained to walking in terrestrial environments, since these environments are often not flat planes, and sometimes can be truly 3D spaces like those many insects encounter when climbing through bushes and trees.

You are right, we have reworded our sentence, removed the contrast made with mantis shrimps and added a reference. Line 50

“Crucially, however, we do not know whether path integration is also possible in aquatic vertebrates such as fish, which, unlike the shrimp, face the additional complexity of navigating with six degrees of freedom of movement 11”

To:

“Crucially, however, we do not know whether path integration is also possible in aquatic vertebrates such as fish, which are nonsurface-bound animals and face a high complexity of navigation with six degrees of freedom of movement 11,12”

-Line 74: Explain why or what you mean when landmark navigation becomes more error prone. Do you mean because landmarks are harder to view and may be misidentified? What do you mean by landmark navigation being slow?

Exactly, landmark navigation will become error-prone because the landmark will be harder to view under turbid conditions and could be misidentified. We gave further details and

added a reference from researchers who have shown that an increase in turbidity leads to a reduction of navigation efficiency and speed in a foraging task (Newport et al. 2021). Lines 77-78

-Line 91: Material in this paragraph is more appropriate in a discussion section.

We followed communications biology journal guidelines that stipulate that “The final paragraph should be a brief summary of the major results and conclusions”. Therefore, after discussion with all co-authors, we wish to keep this paragraph here. We think that the information given is in adequation with the journal guidelines and is relevant for a better understanding by the reader.

Paragraph: “Our results showed that cichlids are capable of path integration, but that they use more than one navigational strategy, including the use of allothetic information and route recapitulation. The unambiguous evidence of path integration implies that the fish had integrated the trajectory from their home shell to the food reward based on the summation of the outward movement vector. This demonstrates that path integration is not limited to terrestrial animals, and that this ability is present in the early vertebrate lineage. Some fish returned to the original position of the shell, demonstrating the use of allothetic place cues (e.g. geometry of the square arena). Finally, evidence of route recapitulation suggests that some fish also learned the outward journey as two consecutive vectors. A substantial proportion of individuals displayed random swimming trajectories, suggesting that they either did not learn any spatial information prior to displacement, or that they lacked motivation to return to their shell. The range of strategies observed is consistent with the hypothesis that navigational strategies are not mutually exclusive; it is likely that it is a combination of possible strategies that ensures navigation is accurate and robust to changing environmental conditions.”

-Line 122: ‘An important number’- state how many fish instead.

We have re-written our sentence and indicated number in brackets. Line 137-138

-Line 141: The level of detail of methods presented here is probably best reserved for the methods section.

After discussion with all co-authors, we wish to keep this paragraph as it stands. An earlier version of the manuscript that included fewer details seems confusing for some of the co-authors. We conclude that this amount of detail was necessary and relevant for a good understanding by the reader. Line 166

Paragraph: “First, 1,000 evenly spaced points were fitted to the fish’s observed trajectory and each of the model trajectories (PI, AC, RR) both cropped to the same size. Then, the Euclidian distance between each pair of interpolated points was extracted (i.e. distance between the tenth interpolated point in the trajectory from the tenth interpolated point in the model trajectory) and used to test if the fish’s trajectory was significantly closer to one of the three model trajectories: PI, AC or RR (Linear mixed model, Table S2, see methods for details). The average distance between the fish trajectory and its closest model trajectory was then compared to 10,000 randomly generated trajectories to determine whether the fish’s trajectory was closer to the model trajectory or non-significantly different from random.”

-Line 231: Since this topic is being discussed, the example in the aquatic environment should be discussed since it is directly relevant to your situation: evidence of landmark and path integration strategies working in concert in mantis shrimp (Patel RN, Cronin TW. 2020. Proc. R. Soc. B 287: 20201898. <http://dx.doi.org/10.1098/rspb.2020.1898>). Another desert ant example is quite a good one for this point as well (Wystrach A, Mangan M, Webb B. 2015. Proc. R. Soc. B 282: 20151484. <http://dx.doi.org/10.1098/rspb.2015.1484>).

Thank you for pointing out these relevant studies, we have added these two references and gave further details about the mantis shrimp experiment in the paragraph starting line 266.

-Line 307: With the current data presented, I think it is a stretch to claim behavioral evidence of path integration in a fish, especially with the strength of how the claim has been written here.

As we have argued, we are confident that our study provides evidence for PI in fish. At the reviewer's request, however, we have revised this sentence (lines 356-357).

-Line 337: How was the arena illuminated?

It was illuminated via fluorescent light following a 12h light/dark cycle. We have added the information lines 388-389.

-Line 546: Where in your manuscript is the visual acuity work discussed besides in the methods? Why was this work done and how does it relate to your work? To make sure the overhead checkerboard was visible? Apologies if I have missed where it is stated.

The visual acuity work was mentioned in the discussion.

We decided to give an estimation of the fish visual acuity to obtain an idea of their ability to discriminate between the squares of the overhead checkerboard pattern (lines 297-298). We have reworded the sentence to make it clearer.

*"Our estimate of the visual acuity of *L. ocellatus* (4.40 CPD, see methods) suggests that they may have been able to see details of the highly contrasted checkerboard pattern 1.5 m above the tank (see Figure 1 in Caves et al. ²⁹ for visual prediction a low CPD). While *L. ocellatus* inhabits depths of up to 20 m in the wild ¹⁶, it is possible that in our experimental aquaria with 0.18 m of water and a depth of 0.45 m (distance to the checkerboard pattern) individuals may have attended to visual information above the water surface."*

I hope you find my comments helpful. I look forward to seeing how your story progresses and I hope to see compelling evidence for path integration in a fish for the first time in the future! Best of luck!

Thank you very much for your relevant comments. We hope that the extra analyses performed as well as the changes made in the figures and in the text will answer your concerns.

** See the Nature Portfolio author and referees' website at www.nature.com/authors for information about policies, services and author benefits

Communications Biology is committed to improving transparency in authorship. As part of our efforts in this direction, we are now requesting that all authors identified as 'corresponding author' create and link their Open Researcher and Contributor Identifier (ORCID) with their account on the Manuscript Tracking System prior to acceptance. ORCID helps the scientific community achieve unambiguous attribution of all scholarly contributions. You can create and link your ORCID from the home page of the Manuscript Tracking System by clicking on 'Modify my Springer Nature account' and following the instructions in the link below. Please also inform all co-authors that they can add their ORCIDs to their accounts and that they must do so prior to acceptance.

If you experience problems in linking your ORCID, please contact the Platform Support Helpdesk.

This email has been sent through the Springer Nature Tracking System NY-610A-NPG&MTS

Confidentiality Statement:

This e-mail is confidential and subject to copyright. Any unauthorised use or disclosure of its contents is prohibited. If you have received this email in error please notify our Manuscript Tracking System Helpdesk team at <http://platformsupport.nature.com> .

Details of the confidentiality and pre-publicity policy may be found here <http://www.nature.com/authors/policies/confidentiality.html>

Reviewers' comments:

Reviewer #1 (Remarks to the Author):

I think the authors have done a very good job at responding to my comments. Moreover, their responses to the other reviewer were very detailed. I think the MS has improved as a result. I'm happy with the new version.

Reviewer #2 (Remarks to the Author):

The authors have made a significant effort to address the concerns of both reviewers during the first round of reviews and should be commended for it. Unfortunately, I am still unconvinced that evidence for path integration, the central point of the paper, is adequately presented in the manuscript. I truly think the additional experiment I previously suggested is necessary to support the authors claims, a recommendation I do not make lightly. I outline specific comments addressing my concerns below. I have included new comments in bold under rebuttals from my previous comments.

2. Grouping the data based on modeled hypothesized data: The swimming trajectories displayed in Figure 3 are grouped by the four groups they were assigned to and ultimately analyzed with. However, a figure of all trajectories compiled together (I've roughly stitched the plots together and included it in the attached PDF) would illustrate the point I am making below. When compiled, all paths appear to create a unimodal distribution of somewhat similar looking paths present in all directions encompassing 90 degrees. Even though the logic for how your trajectories were separated into your four groups is understood, the rationale behind why the data should be segregated is lost. Perhaps a test to show that your data better fits a multimodal rather than unimodal distribution when taken together would justify separating the data into groups. By separating groups without a real justification for doing so, cherry picking may result as an unintended consequence. The 'CircMLE' package in R may be useful to determine if your data fits specific orientation distributions, at least for your angular data, which might be able to help you determine if your data better fits a unimodal or multimodal distribution. Here is the paper that describes how the package can be utilized: Fitak, R. R. and Johnsen, S. (2017) Bringing the analysis of animal orientation data full circle: model-based approaches with maximum likelihood. *Journal of Experimental Biology* 220: 3878-3882.

We understand your concern as we also spent a lot of time evaluating the best methods to analyse these data. Both the authors of the circular package **Dr. Ulric J Lund** and a

Specialist of Monte Carlos analysis **Dr Olivier Bertrand**, were contacted to make sure that we were not missing anything in the analysis.

- 1) **Dr Lund recommended that we not use the circular package for our data, given that our data is restricted to 0-180°. Dr Lund recommended that we compute the circular mean direction as a descriptive statistic, but beyond that, he recommended that we perform ordinary statistics for our data.**

Indeed, in our case, we do not have the problem of 355° being closer to 5° than 330°. In our case 180° is the further angle from 0° and angles are linearly further apart.

We agreed with Dr Lund and performed ordinary statistics that are robust and powerful to analyse our data.

We also used circular statistics to compute the circular means for each group and measure the strength of the uniformity with a Rayleigh test (see Figure 2). We also decided to analyse our data with the package that you mentioned (see 3rd point of this discussion) as a way to support our grouping method.

- 2) **Moreover, we tested grouping options using MC3 analysis with the help of Dr Bertrand.**

The classification on frechet and DTW similarity measures was used on the cichlids trajectories. From this analysis, the optimal number of path clusters was 4 and this was significantly different from 1 (i.e. no cluster). As you can see on the figure below.

The classification was greatly influenced by the endpoint of the cichlid. Since both metrics compare every point in trajectory with another trajectory, the distance increases greatly when the two endpoints are far apart (see below).

To minimize this impact, the trajectories were truncated to the smallest distance travelled by the fish. After repeating the test, we still obtained 4 clusters, separated from left to right (also probably driven by the endpoint).

For this reason, Dr Bertrand recommended we not use this method to classify our trajectories but agreed that multiple clusters are better than none.

3) We tested the package that you suggested (`circ_mle`) to analyse our data.

We would like to thank you for hinting at this paper as it was a circular analysis method and a package that we were not aware of before reading your comments. We feel that it has helped clarify and improve the global analysis.

This package appears to provide more accurate results and a higher level of detail than the simple Rayleigh test from the circular package (Fitak & Johnsen, 2017). **Before discussing our circular analysis, we want to note that our data ranges from 0 to 180°, therefore, circular statistics on these data should be interpreted with care.**

After reading the paper you suggested, we ran a `circ_mle` test on our data. We preliminarily transformed to circular data using the information provided in the paper. We decided to change modulo to π compared to 2π used on the paper as our data can only spread from 0 to π .

```
dc<-circular(Angle.first.straight.path, units = "radian", modulo = "pi").
```

We obtained the following output from `circ_mle(dc)` and the plot function:

	M5A	M2A	M2C	M2B	M3B	M4B	M5B	M4A	M3A	M1
Params	4	2	3	2	3	4	5	3	2	0
$\phi 1$	0.61	1.87	2.11	2.13	5.26	5.43	6.28	5.11	2.27	NA
$\kappa 1$	4.63	1.98	3.77	4.18	0.00	0.00	5.95	2.00	1.82	0.00
λ	0.25	1.00	0.75	0.50	0.50	0.35	0.40	0.25	0.50	1.00
$\phi 2$	2.19	NA	NA	NA	8.41	8.57	2.23	8.25	5.41	NA
$\kappa 2$	4.63	0.00	0.00	0.00	4.54	5.00	5.06	2.00	1.82	0.00
Likelihood	42.63	47.13	50.24	52.32	52.35	52.06	51.18	55.08	65.21	68.00
Convergence	0	0	0	0	0	0	0	0	0	0
AIC	93.26	98.26	106.48	108.64	110.70	112.12	112.36	116.16	134.41	136.00
AICc	94.51	98.61	107.21	109.00	111.42	113.37	114.30	116.88	134.77	136.00
BIC	99.70	101.48	111.31	111.87	115.53	118.57	120.42	120.99	137.64	136.00
Δ AIC	0.00	5.00	13.22	15.39	17.44	18.87	19.11	22.90	41.16	42.75
Δ AICc	0.00	4.10	12.70	14.49	16.92	18.87	19.79	22.38	40.26	41.50
Δ BIC	0.00	1.78	11.61	12.17	15.83	18.87	20.72	21.29	37.94	36.30

Relative likelihoods	1.00	0.08	0.00	0.00	0.00	0.00	0.00	0.00	0.00	0.00
AIC weights	0.92	0.08	0.00	0.00	0.00	0.00	0.00	0.00	0.00	0.00
Evidence Ratio	NA	1.22E+01	7.43E+02	2.19E+03	6.12E+03	1.25E+04	1.41E+04	9.38E+04	8.65E+08	1.92E+09

Figure 1: Raw plot from the circ_mle function. The original parameters from the plot have been left intact meaning that the angular values are in radians units and the values are plotted counter clockwise.

According to this test, and following the model selection procedure based on AIC, the best-selected model for our data is model M5A = homogeneous bimodal ($\Delta AIC=0$, see attached figure, and supplementary material).

The same test performed on the angle from the start of the trajectory to the end of the trajectory led to the selected model M5B=Bimodal ($\Delta AIC=0$, code output table and figure, available on the supplementary material)

We also ran this test with modulo=2pi, to evaluate if it led to any differences. In this case, the default model selection procedure (AIC) selected model M3A= Homogeneous symmetric bimodal, as the best ($\Delta AIC=0$, code output table and figure, available on demand).

A further circular analysis performed to answer your concern #6, lead us to run a non-parametric Watson-Wheeler circular test for homogeneity of means on two or more samples of circular data. We used the Watson-Wheeler circular test for our four groups (PI,APC, RR and Random). The output of this test ($W = 24.77$, $df = 8$, $p\text{-value} = 0.002$) indicates that our groups were significantly not homogeneous.

The results from the discussion with Dr Lund, Dr Bertrand and the latest "circ-mle" analysis performed, led us to conclude that our data do not follow a unidirectional or unimodal distribution (circular analysis) and that grouping can be applied and is significantly better for our data (MC3 analysis). This additional circular data analysis adds strength to our original conclusions. We believe our original grouping method (smallest distance to one of the model trajectories and comparison to 10,000 random trajectories) is relevant and leads to the most conservative results (ensuring that some of the fish tested are certainly following the model trajectory defined instead of showing random movements).

We have decided to add the circular analyses that you recommended in the methods (lines 455 to 466) and results (lines 150-159) with all tables and figures given in supplementary material (Tables S2a and S2a). This will provide the reader with information about the no-unimodal and no-unidirectional distribution of the data to support our grouping methods.

Thank you for working hard to address my concerns and incorporate my comments. Here are my further concerns:

-I still do not see the rationale for separating your data into three groups and then into a further fourth group afterwards. Doing so creates very tight groupings that by definition will match your predictions. Therefore, analyses within these very constrained groups will not tell much as they do not reflect the structure of the data as a whole.

-As for your CircMLE analyses, if a bimodal distribution fits your data best, shouldn't all your data only be segregated into two groups, perhaps using the dashed means presented in the rose diagram above rather than 3 groups which are then further separated into 4?

-Also, if you have a good reason not to use circular analyses since your fish are constrained by the geometry of your reward chamber, I guess you should be consistent with not using circular methods as you state they are improper to use for your situation... Unless you believe they have full 360 degrees of movement once they leave the chamber, in which case you should only use circular methods? Perhaps you should analyze lateral and diagonal displacements separately due to the constraints of each experimental design.

-The orientations of all individuals placed in an orientation plot where each trajectory orientation is a single point on the circumference of the circle (you can make this using the 'circular' package in R) would be enlightening to observe the distribution of the 37 paths you used in your analyses together, separated in groups by color on one plot.

We have re-drawn Figure 2 and included a single point on the circumference of the circle for each individual orientation. Line 172

-What is the rationale for separating the random vs model fitting trajectories into two different plots in Figure 2? They should be presented together as the data was collected together. You can stack the points along the circumference of the plot if you are concerned the reader will not be able to see all the points.

-Since fish were grouped into each hypothesized group based on the model they were most aligned with, it seems obvious that they would be oriented in the expected modelled direction. In this case, the criterion that groups the data, which is quite strict with the addition of further pulling unideal trajectories in each hypothesis group

into a random movement group, and the test of matching the criterion seem quite similar. This highlights the concern of unintended cherry-picking.

The analyses followed two steps that we believe are crucial to obtain appropriate results and an integrated picture of the navigational behaviour of our model species. **First**, we determined if the observed trajectory was significantly closer to one of the model trajectories. **Observed trajectories were grouped by distance to the model trajectories.** Once the trajectory has been categorised within a certain model trajectory we performed an analysis of this distance against 10,000 randomly generated trajectories, allowing us to reject trajectories that could have been following random movement. However, this step did not allow us to determine how well the observed trajectories matched the model expectation. Therefore, **secondly**, we checked if the grouped trajectories fit the expectations of the model trajectory. **We tested angles and lengths.** This allowed us to determine how close the groups matched our hypotheses. If the original grouping made in the first instance was wrong this step would have allowed us to detect it (with $p < 0.05$, for all groups).

This step also allowed us to observe that the fish group categorised under route recapitulation followed the expected model orientation for the first angle and not for the angle start-end. This could be explained by the fact that the fish only retraced the first part of the L-shaped tunnel before orienting back toward the reward chamber. Line 212-213.

-This methodology seems to be designed to filter the best fitting paths out to make conclusions for only those paths. Otherwise, the rationale behind choosing these methods are lacking.

3. For the allothetic cue group, lateral and diagonal displacement groups were analyzed in different groups while both experimental conditions were pooled for the path integration and route recapitulation groups. I believe that lateral and diagonal displacement groups should be separated for all conditions. This is due to the fact that geometries of the arenas are different between the two experimental conditions. Even though your hypothesized path for path integration is the same in both experimental conditions, animals might behave in different ways in the two conditions. This is because animals often use multiple navigational strategies in concert when relevant information is available. These strategies, while separate, are not mutually exclusive. Experiments from desert ants (Wystrach A, Mangan M, Webb B. 2015. Proc. R. Soc. B 282: 20151484. <http://dx.doi.org/10.1098/rspb.2015.1484>) and mantis shrimp (Patel RN, Cronin TW. 2020. Proc. R. Soc. B 287: 20201898. <http://dx.doi.org/10.1098/rspb.2020.1898>) offer a couple examples where multiple navigational strategies are used together. In the lateral displacement experiments, the walls of the arena offer potentially more navigationally relevant information. If you do pool them, in Figure 3 you should plot trajectories of lateral and diagonal displacements independently. If they form a unimodal distribution statistically, then some argument can be made for grouping both experiments together, though it would still be nice for the reader to be able to see this information. Sample sizes should be reported per experimental group as well in table 2. In Figure 3, I would add plots of all trajectories not separated by models but colored by the model group you've assigned them. It would be nice if you could indicate which of

the two experimental conditions they belong to. Perhaps one of the conditions would have dashed trajectories.

We grouped the trajectories for path integration and route recapitulation after both lateral and diagonal displacement because our predictions for model trajectories for path integration and route recapitulation were the same independently of the displacement. We agree with you that following the different displacements, individuals might behave differently because of the external information available and we have therefore done another analysis as you suggested. We used the same previously recommended R package. We tested if individual groups under the path integration group and the route recapitulation group followed a unimodal or bimodal distribution.

The best model that described the path integration trajectories was M2A= Unimodal distribution ($\Delta AIC=0$, see table and figure below).

	M2A	M2C	M2B	M4A	M3B	M5B	M5A	M4B	M3A	M1
Params	2	3	2	3	3	5	4	4	2	0
$\phi 1$	2.13	2.05	2.16	5.37	2.17	2.53	5.78	5.16	2.13	NA
$\kappa 1$	4.67	3.86	5.42	3.08	5.38	3.02	4.00	5.00	4.64	0.00
λ	1.00	0.75	0.50	0.25	0.50	0.55	0.25	0.25	0.50	1.00
$\phi 2$	NA	NA	NA	8.52	5.31	1.74	2.30	8.30	5.28	NA
$\kappa 2$	0.00	0.00	0.00	3.08	0.00	2.04	4.00	2.66	4.64	0.00
Likelihood	5.73	7.29	8.79	8.49	8.79	7.17	8.48	8.83	11.27	14.70
Convergence	0	0	0	0	0	0	0	0	0	0
AIC	15.45	20.58	21.58	22.99	23.59	24.35	24.96	25.66	26.54	29.41
AICc	17.85	26.58	23.98	28.99	29.59	54.35	38.30	38.99	28.94	29.41
BIC	15.61	20.82	21.74	23.22	23.82	24.75	25.28	25.98	26.70	29.41
ΔAIC	0.00	5.13	6.13	7.53	8.13	8.90	9.51	10.21	11.08	13.95
$\Delta AICc$	0.00	8.73	6.13	11.13	11.73	36.50	20.44	21.14	11.08	11.55
ΔBIC	0.00	5.21	6.13	7.61	8.21	9.13	9.67	10.36	11.08	13.80
Relative likelihoods	1.00	0.08	0.05	0.02	0.02	0.01	0.01	0.01	0.00	0.00
AIC weights	0.84	0.06	0.04	0.02	0.01	0.01	0.01	0.01	0.00	0.00
Evidence Ratio	NA	12.98	21.39	43.24	58.34	85.44	116.21	164.42	255.11	1071.48

Due to this result, we have decided to keep PI trajectory data pooled together independently of the displacement followed by the fish, and we did similarly for the RR group.

- We have added the sample size for individuals following lateral and diagonal displacement in Table 1.
- We have added the sample size per experimental group in Table 2.
- We changed Figure 3, all trajectories are now presented on the same plot and coloured by the model group. The two experimental conditions (lateral and diagonal displacement) can be differentiated by the line type (plane versus dashed).

Does Figure 3 include all paths from both lateral and diagonal displacements? The reason I ask is that shouldn't a black model line for allothetic place cues for diagonal displacements also be included? The bounds of the box in (B)

also indicate that only lateral displacements are presented even though I am fairly confident this is not the case. Do no fish from diagonal displacements perform expected movements for allothetic place cues?

5. I know additional behavioral experiments are a lot of work and are a lot to ask for. Therefore, I am generally very hesitant to ask for additional experiments. However, in this case I believe an additional experiment is necessary to alleviate some of my concerns. I suggest you replicate the diagonal displacement experiment exactly with one change: The chamber should have the door open on the opposite side from where the door currently is (perhaps you can have a door on each side of the chamber). You would then be able to see if fish leave the displacement chamber in a randomly oriented fan constrained to approximate 90 degrees in the direction of the door, mirroring the results presented in your current manuscript and supporting my concerns, or swim around the chamber to the location predicted from PI, giving your interpretation of your current results much more validity and giving you stronger results to report in an updated manuscript. For clarity I've attached an illustration of this experiment viewable in the attached PDF.

Suggested Experiment

We think that this extra experiment is not necessary for the following reasons:

- 1) The major reason is the non-unimodal distribution of the fish trajectories. From this non-uniform dataset, an APC, a RR and a PI pattern could be clearly distinguished from the random using appropriate stats.
- 2) Once the door is open the fish had 180-degree freedom of movement, allowing them to follow multiple navigational cues. Our experimental paradigm rigorously considered all different possible scenarios in fish navigational strategies and tested them against random movement. We were able to clearly separate the three different scenarios and used a very conservative cut-off to group our fish (see later point 4).
- 3) The displacement has been made in two different positions. This gave the fish access to a different view once they had exited the chamber. The fish did show PI trajectories (and the other model trajectories) in these two displacements. We think that this is a strength of our experiment and analyses. In most path integration displacement experiments the individuals were displaced to a single location.

- 4) Moreover, thanks to the two possible displacements, the inverse of flipping the door around 180 to check for PI trajectories was essentially done with respect to the allothetic trajectories (APC, figure 1d). The fish had to follow different trajectories to perform APC strategies after the lateral or diagonal displacements (the trajectory they had to follow for APC strategy after diagonal displacement is the mirrored trajectory that the one you suggest here for an individual that would follow PI if the door was opened on the back of the chamber). Our result showed that fish followed the APC strategy after the two displacements and therefore indicates that fish would be able to perform PI trajectory in another configuration.
- 5) Our statistical analysis was purposely conservative to ensure that the fish that followed PI trajectories, or the other model trajectories, were not performing random behaviour. We hope that the extra analyses that we detailed above have assured you and that you agree with our interpretations.
- 6) Ethical consideration: We followed the 3Rs guidelines in all our experiments and have used the minimum adequate number of individuals allowing us to answer our research question. Running further experimental trials would require using an important number of extra individuals (as detailed in the manuscript the participation rate to the experiment was low). We think that this would be against the 3Rs recommendations.

For the reasons listed above, we believe that extra experiments would not actually add more information to our current results. From a practical point of view, changes to our facilities since the experiment was performed in 2021 also mean we would be unable to replicate the experiment under the exactly same conditions.

I unfortunately still believe that experiment I originally suggested should be done to alleviate the concerns I bring up. I address your numbered rebuttals point by point:

-For rebuttal points 1 and 2, the rationale for these groupings are still not clear. Overly complex analyses seem to be compensating for flaws in experimental design.

-Point 3- Again evidence of path integration is not convincingly shown. Thus, the recommendation of the above experiment.

-Point 4- Firstly, APC and PI are two different navigational strategies. Thus, even if convincing evidence of APC was present (which I believe is lacking in the current study), that would not apply that confirmation to evidence of a different navigational strategy (PI) being present as well.

Second, evidence of APC occurring during diagonal displacement experiments are not seen (at least from what is presented in Figure 3, see above comment).

-Point 6- For ethical reasons, I agree that unnecessary experiments should not be done, even non-invasive behavioral experiments such as the ones presented here. But one may argue that it is even more unethical to use animals for experimentation without completing experiments required to adequately provide evidence for the main theses of the paper, as the animals used in previous trials would be a waste.

-Final points- Surely the experimental situation– a square freshwater tank illuminated by a fluorescent bulb– could be replicated to some degree of closeness. I understand that in practical terms, extra experiments are a lot to

ask for. But I cannot agree the current study has shown that these fish are performing path integration. The biggest messages I take from the work are that the cichlids here *might* be performing path integration, allothetic place navigation, route recapitulation, and random undirected movements together in this assay, but all these options at this point are hypotheses generated by your results. It is my opinion that you need to perform more rigorous experiments testing the hypothesis that these fish are performing path integration in order to appropriately make the definitive statements written in lines 20, 95, 245 and 356, including ‘unambiguous evidence’ and ‘demonstrating for the first time that a teleost species can perform path integration’. The simplest and way to do this with your current setup and the work you present here would be by running the experiment I originally suggested, in my opinion. During this experiment, if a group of animals travelled around the reward chamber to the expected path integration location, I would be far more convinced that what is observed in your current data aren’t random movements as a whole. If you are unwilling to run additional experiments that more rigorously test the path integration hypothesis, you should reform the central thesis of your paper to instead make clear that your fish might be performing RR, PI, APC, and random movements, but further work is necessary to definitively state that these navigational strategies are being utilized in your assay. I commend you for your efforts so far and really do hope you convincingly show path integration in a fish for the first time. I would be very excited to read over such work!

-Some readers might be interested in seeing the goodness of fit metrics for each fish’s path to the three original models and the random model to compare how much better they fit to one model than another. I’m not sure if the distances reported in Table S2 truly clarify this question. For example, in Table S2, the bolded lowest difference indicates what group each trajectory was placed. But the Means plus/minus SDs of the lowest deviating model distance often completely encompass the distances of the other models, making it seem as if there isn’t a good reason to assign the trajectory to a specific model. Perhaps AICs or BICs would be useful for readers to evaluate this?

We have added AIC and BIC values to Table S3 (which was table S2 in the original submission, see supplementary material).

We choose to present SD error bars which are descriptive error bars (Cumming et al. 2007- Errors bars in experimental biology). These error bars indicate the average difference between the data point and their mean, and should not be used to predict statistical significance (Cumming et al. 2007). We could use inferential error bars such as SE or CI if you want us to make the change (SE are actually small and SE around each mean do not overlaps). However, we choose to have SD presented for all means here as random-SD_{uniform} for the criteria to categorise a trajectory into one of the model trajectories or to random movement trajectory.

-I think deltaAICs and deltaBICs between the current and runner up models would be more useful than just AIC and BIC values to evaluate model fit.

New Comment: Line 246- 'This shows that vector-based navigation is not limited to terrestrial species'. As you have previously cited, this has already been demonstrated.

Reviewer #3 (Remarks to the Author):

1. The question of this study (do fish path-integrate) is obviously an interesting one and the methodology seems fine and the choice of fish species excellent. One worry is that the fish swim in a 3-D tank, but just the horizontal component of the paths are recorded. Results might become clearer with a lower water level that is just sufficient for the fish to swim.

2. Despite the elaborate statistics, It is hard to be convinced that there's sufficient data to conclude that fish can path integrate. If I've understood Table 1 correctly, there are just 8 out of 37 attempts in which PI occurred. In Fig. 2. It doesn't look as though the endpoints of the PI group are much more clustered than the random group.

3. It seems worth attempting to improve the success rate. One way forward may be to experiment with a lower water level on a few fish. If you use a variety of shapes and lengths of outward tunnels, you can get multiple data points from one animal. It could be that the fish have to become accustomed over several trials to the strange situation of swimming through a tunnel and being trapped. And that only after this experience are they are settled enough to perform PI with a newly shaped outward tunnel.

Response to reviewer

Reviewer #3 (Remarks to the Author):

1. The question of this study (do fish path-integrate) is obviously an interesting one and the methodology seems fine and the choice of fish species excellent. One worry is that the fish swim in a 3-D tank, but just the horizontal component of the paths are recorded. Results might become clearer with a lower water level that is just sufficient for the fish to swim.

In the experimental tank, the fish could swim in 12 cm deep water. Moreover, both the shell (which was either on or buried in the sand) and the food reward (provided at the bottom of the reward chamber) were located in the same lower horizontal plane. Therefore, we rarely observed fish moving in the water column. The vertical movement was relatively restricted in the experimental apparatus, compared to the fish home tank (32 cm water depth) or their natural environment (Lake Tanganyika). This depth was also chosen in accordance with the ethics committee to prevent the possibility of fish being stressed due to a low water level.

We are currently setting up a follow-up experiment, with another species, to test the precision of fish in a three-dimensional environment (79 cm depth) compared to a two-dimensional environment (shallow water). In this planned experiment, we will be able to provide food rewards at different heights in the water column and test the accuracy of navigation in the three dimensions.

Although the fish had some possibility to move vertically, our conservative analyses show evidence consistent with some individual cichlids using path integration strategy. We have also revised our manuscript and present a more cautious interpretation of our data.

2. Despite the elaborate statistics, It is hard to be convinced that there's sufficient data to conclude that fish can path integrate. If I've understood Table 1 correctly, there are just 8 out of 37 attempts in which PI occurred. In Fig. 2. It doesn't look as though the endpoints of the PI group are much more clustered than the random group.

Our study aimed to determine if fish can path integrate and not if they all do it. Fish are likely to use a range of navigational strategies that could be dependant on the cues available in their environment at a given time, but also of intrinsic factors (age, sex...).

The low engagement of the fish has limited our ability to obtain as much data as we wanted when we started the experiment. However, it highlights the attachment of male *L. ocellatus* to their shells, which was a prerequisite for testing path integration. Moreover, we applied a conservative analysis that has prevented us from categorising fish in the wrong navigational strategy (or random behaviour).

The endpoints of the PI group are more clustered than the random group and were not significantly different from a 45-degree angle ($V=32$, $p=0.055$). Two individuals showed a longer homing trajectory, with a final angle that deviated more than the other fish. This could be because of error accumulation which increases with the length of the trajectory. Such results are commonly observed when individuals perform path integration.

3. It seems worth attempting to improve the success rate. One way forward may be to experiment with a lower water level on a few fish. If you use a variety of shapes and lengths of outward tunnels, you can get multiple data points from one animal. It could be that the fish have to become accustomed over several trials to the strange situation of swimming through a tunnel and being trapped. And that only after this experience are they settled enough to perform PI with a newly shaped outward tunnel.

The reviewer's suggestion is an interesting and valid follow-up study but represents a separate experiment that is beyond the scope of the current study. The idea of accustoming the fish to a tunnel and then changing the tunnel shape and testing the fish on multiple shapes is indeed likely to improve the success rate. We will consider this information for future investigations. However, we would need to implement additional controls to ensure accurate results. Path integration relies on untrained behaviour, so training fish to navigate through a restricted tunnel for experimental purposes may not be the best approach. This is because we are uncertain how habituation to navigation in a restricted shape (over a high number of training trials) would affect vector computation once the fish are tested.